# Interspike interval correlations in neuron models with adaptation and correlated noise

**Lukas Ramlow**[1,2]*, **Benjamin Lindner**[1,2]

**1** Bernstein Center for Computational Neuroscience Berlin, Berlin, Germany, **2** Physics Department, Humboldt University zu Berlin, Berlin, Germany

* lukas.ramlow@bccn-berlin.de

**Data Availability Statement:** All source code and data files are available from the open science framework database at: https://osf.io/yj389/?view_only=67665f4c961d496c991db3cfc69fc4c1.

## Abstract

The generation of neural action potentials (spikes) is random but nevertheless may result in a rich statistical structure of the spike sequence. In particular, contrary to the popular renewal assumption of theoreticians, the intervals between adjacent spikes are often correlated. Experimentally, different patterns of interspike-interval correlations have been observed and computational studies have identified spike-frequency adaptation and correlated noise as the two main mechanisms that can lead to such correlations. Analytical studies have focused on the single cases of either correlated (colored) noise or adaptation currents in combination with uncorrelated (white) noise. For low-pass filtered noise or adaptation, the serial correlation coefficient can be approximated as a single geometric sequence of the lag between the intervals, providing an explanation for some of the experimentally observed patterns. Here we address the problem of interval correlations for a widely used class of models, multidimensional integrate-and-fire neurons subject to a combination of colored and white noise sources and a spike-triggered adaptation current. Assuming weak noise, we derive a simple formula for the serial correlation coefficient, a sum of two geometric sequences, which accounts for a large class of correlation patterns. The theory is confirmed by means of numerical simulations in a number of special cases including the leaky, quadratic, and generalized integrate-and-fire models with colored noise and spike-frequency adaptation. Furthermore we study the case in which the adaptation current and the colored noise share the same time scale, corresponding to a slow stochastic population of adaptation channels; we demonstrate that our theory can account for a nonmonotonic dependence of the correlation coefficient on the channel's time scale. Another application of the theory is a neuron driven by network-noise-like fluctuations (green noise). We also discuss the range of validity of our weak-noise theory and show that by changing the relative strength of white and colored noise sources, we can change the sign of the correlation coefficient. Finally, we apply our theory to a conductance-based model which demonstrates its broad applicability.

**Funding:** LR and BL received funds of Deutsche Forschungsgemeinschaft (http://www.dfg.de): LR received funds of Deutsche Forschungsgemeinschaft: LI-1046/4-1. The funders had no role in study design, data collection and analysis, decision to publish, or preparation of the manuscript.

**Competing interests:** The authors have declared that no competing interests exist.

### Author summary

The elementary processing units in the central nervous system are neurons that transmit information by short electrical pulses, so called action potentials or spikes. The generation of the action potential is a random process that can be shaped by correlated fluctuations (colored noise) and by adaptation. A consequence of these two ubiquitous features is that the successive time intervals between spikes, the interspike intervals, are not independent but correlated. As these correlations can significantly improve information transmission and weak-signal detection, it is an important task to develop analytical approaches to these statistics for well-established computational models. Here we present a theory of interval correlations for a widely used class of integrate-and-fire models endowed with an adaptation mechanism and subject to correlated fluctuations. We demonstrate which patterns of interval correlations can be expected from the interplay of colored noise, adaptation and intrinsic nonlinear dynamics.

## Introduction

Neural activity or spiking is a stochastic process due to the presence of multiple sources of noise, including thermal, channel, and synaptic noise [1]. The study of neural systems in terms of stochastic models is hence vital for understanding spontaneous neural activity as well as neural information processing. Particularly useful in this respect are integrate-and-fire (IF) models [2–4] because these models are often analytically tractable and thus permit insights into the interplay of noise, signals, and nonlinear neural dynamics. It should be also noted that they can mimic the neural response to in-vivo-like inputs for some cells surprisingly well [5–7] and there exist procedures to systematically map biophysically detailed conductance-based models to this model class (see e.g. [8]).

A common simplification in the study of neural spike generators lies in the assumption that times between subsequent spikes, the interspike intervals (ISIs), are statistically independent. Put differently, neural spiking is assumed to be a renewal process [9], which allows for a far-reaching theory of neural interactions in recurrent networks [10, 11]. We note that simple (one-variable) IF neurons, if driven by uncorrelated fluctuations, will exactly generate such a renewal spike train and for this reason lots of theoretical efforts have focussed on the problem of calculating the ISI probability density (statistics that completely characterizes a renewal process) [2, 12, 13].

Although renewal theory has been successful in describing some aspects of neural activity, there is increasing experimental evidence that in many cases ISIs are correlated over a few lags [14–23]. These correlations are an important statistics of spike trains as they shape spectral measures and therefore have consequences for information transmission and signal detection [15, 16, 21, 24–27].

Such correlations can be quantified by the serial correlation coefficient (SCC)

$$\rho_k = \frac{\langle (T_i - \langle T_i \rangle)(T_{i+k} - \langle T_{i+k} \rangle) \rangle}{\langle (T_i - \langle T_i \rangle)^2 \rangle} \tag{1}$$

where $T_i$ is the $i$th ISI, $k$ represents the lag and $\langle \cdot \rangle$ denotes the ensemble average. The SCC $\rho_k$ measures whether two intervals' deviation from the mean are on average proportional ($\rho_k > 0$), anti-proportional ($\rho_k < 0$) or independent of each other ($\rho_k = 0$).

Positive correlations can be induced by correlated input due to synaptic filtering [28, 29], slow network processes [30–32] or channel noise with slow kinetics [33, 34]. Another mechanism, commonly associated with negative ISI correlations, exhibited by many neurons is spike-frequency adaptation, i.e. the increase of the ISI following an initial decrease due to a stimulation. Adaptation currents include calcium-gated potassium currents, M-Type currents as well as the slow recovery of sodium channels [35] (for the computational role of these and other neural adaptation mechanisms, see [36]). Typical time scales of these currents range from 50ms to 1s and can therefore by far exceed the mean ISI. Interestingly distinct causes of correlations can result from a single source: In neurons of the sensory periphery adaptation-channel noise, i.e. the stochastic opening and closing of slow ion channels that mediate an adaptation current, may dominate the spiking statistics and provide at the same time adaptation and correlated fluctuations [33, 34].

While both correlated input, in the form of colored noise, and adaptation and their implications for ISI correlations have been studied separately [37–39], a general theory that allows to calculate the SCC in the presence of multiple correlation-inducing processes is still missing. In this article we extend the *weak* noise theory developed by Schwalger and Lindner [39] for mean-driven neurons to include multiple correlation-inducing processes. Our theory mainly applies to noisy neurons in the sensory periphery, in which the type of noise and the adaptation mechanisms are known. Cortical neurons, on the other hand, are more difficult: they typically operate in an excitable firing regime and the network noise that drives them is generally not known with respect to its statistics; below we discuss a special case of cortical firing that can nevertheless be captured by our theory.

We relate statistics of the spike train, namely the SCC $\rho_k$ to intrinsic properties of nonlinear neural dynamics captured by the phase-response curve (PRC). The PRC measures the shift of the next spike time of a neuron subject to a small perturbation at different times in the firing cycle [40–42]; see Fig 1 for an illustration of the method. The shape of the PRC depends crucially on the neuron type: type 1 neurons which bifurcate from a quiescent to a tonically firing regime via a saddle-node on invariant circle bifurcation possess a purely positive PRC, whereas

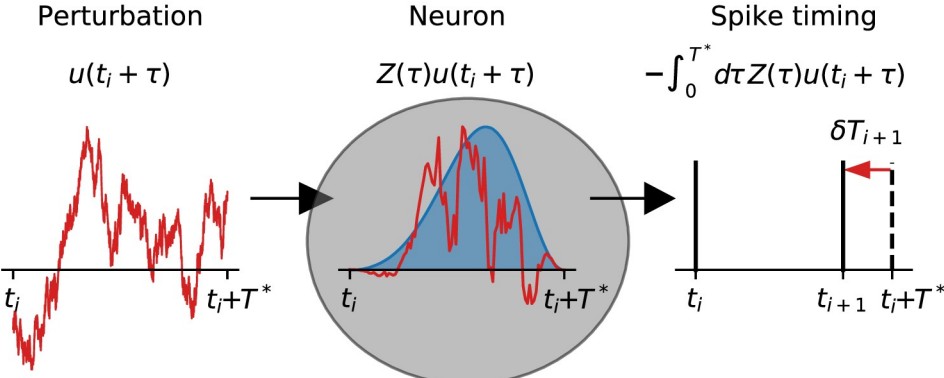

**Fig 1. Linear response of a neuron models spike timing.** The membrane potential of a tonically firing neuron with deterministic ISI $T^*$ is subject to an arbitrary perturbation $u(t_i + \tau)$ (left panel, red line). Here $t_i$ is the time of a reference spike and $\tau \in [0, T^*]$ is the relative time since the last spike, resembling a phase. How strongly this perturbation advances or delays the "phase" $\tau$ will in general depend on the "phase" itself at which the perturbation is applied. This sensitivity is quantified by the phase-response curve $Z(\tau)$ shown in blue in the middle panel. The term $Z(\tau)u(t_i + \tau)$ can thus be thought of a perturbation of the phase (middle panel, red line). In linear response these phase perturbations can be integrated to yield the cumulative phase shift or spike time derivation $\delta T_{i+1} = t_{i+1} - (t_i + T^*)$ (right panel, red arrow). Note that separating the perturbation from the deterministic dynamics of the neuron model, i.e. finding $u(t_i + \tau)$ is part of the problem that we address in this paper and solve in detail in the Methods section.

type 2 neurons which undergo a a supercritical Hopf bifurcation have a partially negative PRC (see [8, 42] for more details on these neuron types). Our theory applies to both neuron types and predicts qualitatively novel patterns of interval correlations that deviate from a single geometric sequence; such deviations have been recently reported experimentally [43, 44].

The paper is organized as follows. We first introduce the broad class of models and the correlation measure of interest; we illustrate both by a special case, the quadratic IF model with adaptation and colored noise. In the following section, we present the general expression of the serial correlation coefficient in terms of the phase-response curve, adaptation kernel, and correlation function of the colored noise. We then explore the role of the specific shape of the phase-response curve on the SCC by considering integrator and resonator models with purely positive and partially negative PRC, respectively. We also discuss the case of a slow population of stochastic ion channels that can be approximated by our model [33]. Finally, we demonstrate that our theory can be applied to a conceptually different model, namely the conductance-based Traub-Miles model with an M current. We conclude our study with a discussion of the results in the context of neural information transmission and give an outlook to several open problems.

## Results

Here we study a stochastic multidimensional integrate-and-fire neuron model with membrane potential $v(t)$ and $N$ auxiliary variables $w_j(t)$ that is subject to spike-triggered adaptation (variable $a(t)$) as well as correlated and uncorrelated Gaussian noise sources $\eta(t)$ and $\xi_v(t)$, respectively.

$$\dot{v} = f_0(v, \mathbf{w}) + \mu - a + \eta + \sqrt{2D}\xi_v(t), \tag{2a}$$

$$\dot{w}_j = f_j(v, \mathbf{w}), \qquad\qquad j = 1, \dots, N \tag{2b}$$

$$\tau_a \dot{a} = -a + \Delta \sum \delta(t - t_i), \tag{2c}$$

$$\tau_\eta \dot{\eta} = -\eta + \sqrt{2\tau_\eta \sigma^2}\xi_\eta(t). \tag{2d}$$

We apply the usual fire-and-reset rule: when $v(t)$ reaches a threshold $v_T$, a spike is triggered at time $t_i = t$; the membrane potential $v$ and the auxiliary variables $w_j$ are instantaneously reset to $v = v_R$ and $\mathbf{w} = \mathbf{w}_R$, respectively. In contrast to that, $a(t)$ undergoes a jump by $\Delta/\tau_a \geq 0$. In the absence of any noise we assume that the system approaches a limit cycle (dashed line in Fig 2B) with a fixed period $T^*$ and a unique value of the adaptation variable right after a spike, $a^*$.

Here we focus on the full stochastic system in which we use the uncorrelated noise sources $\xi_v(t)$, $\xi_\eta(t)$, independent zero-mean Gaussian white noise processes with $\langle \xi(t)\xi(t')\rangle = \delta(t' - t)$. As a consequence, $\eta(t)$ represents a temporally correlated (colored) Ornstein-Uhlenbeck (OU) process with auto-correlation function $\langle \eta(t)\eta(t')\rangle = \sigma^2 \exp(-|t' - t|/\tau_\eta)$, i.e. Eq (2d) is a Markovian embedding for low-pass filtered noise (for more general embedding of colored noise in IF neurons, see [45]). The presence of both white and colored noise will affect the voltage dynamics directly. However, there is also an indirect effect through noise-induced deviations of the adaptation variable from the deterministic limit cycle. As outlined in the method section, the combined effect of the direct and indirect perturbations on the next spike time is subsumed in the perturbation function $u(t)$ that measures the deviation from the deterministic limit cycle and is shown exemplary in Fig 1 (for the detailed definition of $u(t)$ see Eq (35)). This function

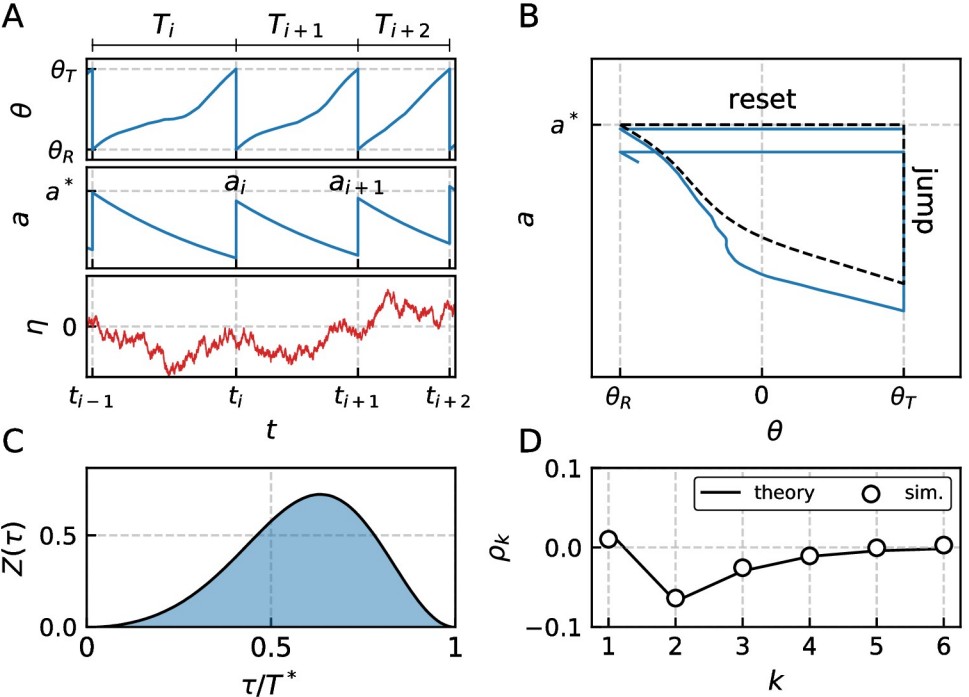

**Fig 2. Serial correlation coefficients for the adaptive QIF (Theta) model with colored noise.** Panel A shows the transformed membrane potential $\theta(t) = 2\tan^{-1}v(t)$, adaptation current $a(t)$ and colored noise $\eta(t)$ with spike times $\{t_i\}$, ISIs $\{T_i\}$ and peak adaptation values $\{a_i\}$. Panel B displays the deterministic limit cycle (dashed line) and exemplary noisy trajectory (solid line) in the phase plane $(\theta, a)$. Note that the jump is of constant size $\Delta/\tau_a$, while the voltage or equivalently phase always resets to a fixed value $\theta_R$. Panel C depicts the corresponding type I PRC quantifying the QIF model's response characteristics. For a non-adaptive QIF model the PRC would be symmetric around $T^*/2$; for the adaptive QIF, however, the maximum is shifted towards the right, i.e. the neuron is particularly sensitive to stimuli applied at the end of the ISI. Panel D shows the SCC with initial, slightly positive correlation coefficient due to positively correlated noise and subsequent negative correlations governed by adaptation. This pattern cannot be described by a single geometric sequence. Parameters: $\mu = 5$, $\tau_a = 6$, $\Delta = 18$, $\tau_\eta = 4$, $\sigma^2 = 0.5$, $D = 0$ and resulting $T^* \approx 4.0$ and coefficient of variation $C_V \approx 0.2$.

typically attains both positive and negative values and carries memory about previous activity and stimuli.

The model is an extension of the one considered by [39]. The crucial novel feature is the addition of a colored noise process. Consequently, here we deal with two possible sources of interspike interval correlations—spike-frequency adaptation and slow fluctuations, as found for instance in the case of adaptation-channel noise [33, 34]. Note that the fixed reset of the voltage and auxiliary variable ensures that the spike-train is renewal in the absence of these two slow processes. In addition, a minor difference of the models is the scaling of the jump amplitude; in [39] the jump amplitude was $\Delta$ and not $\Delta/\tau_a$ but the latter choice is the natural one for systems with adaptation-channel noise.

While our general theory does not impose any restriction on the dimensionality of the chosen IF model we will discuss only a selection of paradigmatic models with one auxiliary variable at most, namely the quadratic IF ($N = 0$), leaky IF ($N = 0$) and generalized IF ($N = 1$) model. An interesting special case, the adaptive quadratic integrate-and-fire model (QIF), is considered in Fig 2. It is a model without auxiliary variables ($N = 0$), a quadratic nonlinearity in the voltage dynamics $f_0(v) = v^2$, and threshold and reset points at infinity $v_T = -v_R = \infty$. The model is the normal form of a type I neuron for which the transition from the excitable regime to the tonically firing regime occurs via a saddle-node bifurcation [8, 42], implying a

non-negative phase-response curve (PRC) [46]. Furthermore, the QIF model is equivalent to the Theta-neuron by the transformation $\theta = 2\tan^{-1} v$ and thus possesses identical statistical properties; in particular, they share identical ISI correlations.

The time courses $\theta(t)$, $a(t)$ and $\eta(t)$ and the limit cycle in the $\theta$-$a$-plane are shown in Fig 2A and 2B, respectively. The central response characteristics of a tonically firing neuron, the PRC $Z(t)$, is displayed in Fig 2C. Our theory, that is based on this PRC and detailed in the following, shows excellent agreement with the numerically simulated SCC (theoretical predictions and simulation results are shown in Fig 2D). The shown pattern is unlike any other one discussed in the theoretical literature on ISI correlations so far: very weak positive correlations $\rho_1$ between adjacent ISIs and pronounced negative correlations at all higher lags $k > 1$. This is due to a non-trivial interplay between adaptation and colored noise. The observed shape of the correlations is only one of several distinct patterns that are possible in our model and explored in the following by means of our analytical approximations.

Generally and in line with our *weak* noise assumption we discuss cases where the noise intensity is rather small. However, in a later Section we explore the range of validity of our theory in terms of the output variability of the spike train and find quantitative agreement up to $C_V \approx 0.2$. Qualitatively, typical correlation patterns (i.e. SCC as a function of the lag) are well described for an even larger values of $C_V \approx 0.7$.

## General expression for the correlation coefficient

As pointed out above, we assume that our model in the absence of noise operates in the tonically firing regime with deterministic period $T^*$ and that the spike train in the presence of noise is a stationary stochastic process. As shown in the Methods section, if the neuron is subject to a weak noise, the interspike intervals will be correlated according to the serial correlation coefficient (SCC)

$$\rho_k = \left(\frac{A}{C}\right)\rho_{k,a} + \left(\frac{B}{C}\right)\rho_{k,\eta}, \quad k > 0 \tag{3}$$

with coefficients

$$A = 1 + \frac{(1 + (\alpha v)^2 - 2\alpha v \beta)}{\alpha v - \beta}\rho_{1,\eta} - \alpha v \beta, \tag{4a}$$

$$B = \frac{(1 - (\alpha v)^2)(1 - \alpha \beta)(\alpha - \beta)}{(1 + \alpha^2 - 2\alpha^2 v)(\alpha v - \beta)}, \tag{4b}$$

$$C = 1 + 2\rho_{1,a}\rho_{1,\eta} - \alpha v \beta \tag{4c}$$

$$\alpha = e^{-\frac{T^*}{\tau_a}}, \quad \beta = e^{-\frac{T^*}{\tau_\eta}}, \quad v = 1 - \frac{a^*}{\tau_a}\int_0^{T^*} d\tau \; Z(\tau)e^{-\tau/\tau_a}. \tag{4d}$$

and deterministic peak adaptation value $a^* = (\Delta/\tau_a)/(1 - \exp[-T^*/\tau_a])$. Remarkably, the dependence on the lag $k$ is carried exclusively by the two specific SCCs $\rho_{k,a}$ and $\rho_{k,\eta}$. The prefactors though depend on properties of both the adaptation as well as the noise sources. The coefficient $\rho_{k,a}$ describes the correlations in the case of adaptation and purely white noise ($\sigma = 0$). The second coefficient $\rho_{k,\eta}$ represents the correlations in the absence of adaptation ($\Delta = 0$) but with the combination of white and colored noise. The specific SCC $\rho_{k,a}$ is identical to the one

derived in [39]:

$$\rho_{k,a} = -\frac{\alpha(1-\alpha^2 v)}{1+\alpha^2 - 2\alpha^2 v}(1-v)(\alpha v)^{k-1}, \tag{5}$$

The SCC $\rho_{k,\eta}$ is derived in the Methods section and reads

$$\rho_{k,\eta} = \frac{\int d\omega |\tilde{Z}(\omega)|^2 (1+\omega^2\tau_\eta^2)^{-1}e^{-i\omega T^*}}{\int d\omega |\tilde{Z}(\omega)|^2 [(1+\omega^2\tau_\eta^2)^{-1} + D/(\tau_\eta\sigma^2)]}\beta^{k-1}, \tag{6}$$

where $\tilde{Z}(\omega) = \frac{1}{T^*}\int_0^{T^*} d\tau Z(\tau)e^{-i\omega\tau}$. Interestingly, Eq (6) is the only place where the noise strength parameters $D$ and $\sigma^2$ enter and they do so as the ratio of noise intensities $D/(\tau_\eta\sigma^2)$. In this formulation it is simple to see that $\rho_{k,\eta}$ vanishes for $D \gg \tau_\eta\sigma^2$. In the opposite limit of vanishing white noise ($D = 0$), Eq (6) coincides with the expression derived in [47].

Our main result Eq (3) implies that the SCC for a general stochastic IF model with both adaptation and correlated noise is in fact a linear combination of two geometric series. These geometric series are determined by the two correlation-inducing processes and agree with the specific SCCs Eqs (5) and (6) except for a constant prefactor (constant with respect to the lag $k$). A sum of two geometric series as in Eq (3) can exhibit completely different patterns of interval correlations compared to a single geometric sequence obtained in previous theoretical calculations [33, 37, 38, 47–49]. We recall that the absolute value of the elements of a geometric sequence $s_k = s_0 r^k$ decay exponentially with the lag $k$ for physically plausible values $|r| < 1$. In addition, the SCC's sign is determined by the prefactor $s_0$ and the sign between adjacent elements may alternate depending on the sign of the base $r$. The two possible signs of $s_0$ and $r$ allow for four distinct patterns of correlations in the case of a single geometric sequence.

Possible shapes that result from the interplay between the specific SCCs are illustrated in Fig 3. The pattern in Fig 3A for instance, is characterized by a very small positive first correlation coefficient (this could also be amplified or diminished by fine tuning parameters) whereas higher lags have pronounced negative correlations—a structure that cannot be generated by a single geometric sequence. In Fig 3B the inverse case is shown: a weak and negative first correlation coefficient followed by stronger positive coefficients at higher lags. Deviations from a single geometric sequence have been seen experimentally (see [44] for a recent example). Indirect evidence for the combination of short-term negative and long-term positive correlations have been reported by means of the Fano factor [50–52] (see [21] for an explanation of the underlying connection between correlations and Fano factor).

A closer inspection of Eqs (5) and (6) reveals that interval correlations introduced by the OU process lead to a coefficient $\rho_{k,\eta}$ that can only decay exponentially with $k$ because for the base of the power $\beta^{k-1}$ we have the condition $0 < \beta < 1$, according to Eq (4d). A richer repertoire, however, becomes possible for $\rho_{k,a}$ because the base of the power can attain values from a broader interval $-1 < \alpha v < 1$ (see Methods); note that $\alpha > 0$ and hence the sign of the base is determined by $v$. Thus oscillatory correlation coefficients enveloped by an exponential function emerge for a negative base, $v < 0$. For $v > 0$ the purely exponential case is recovered.

In addition to the base, also the prefactor can attain different signs which specifically depends on the neuron's PRC. In the following sections we discuss patterns of interval correlations for two distinct cases, that is the leaky IF model with a non-negative and the generalized IF model with a partially negative PRC.

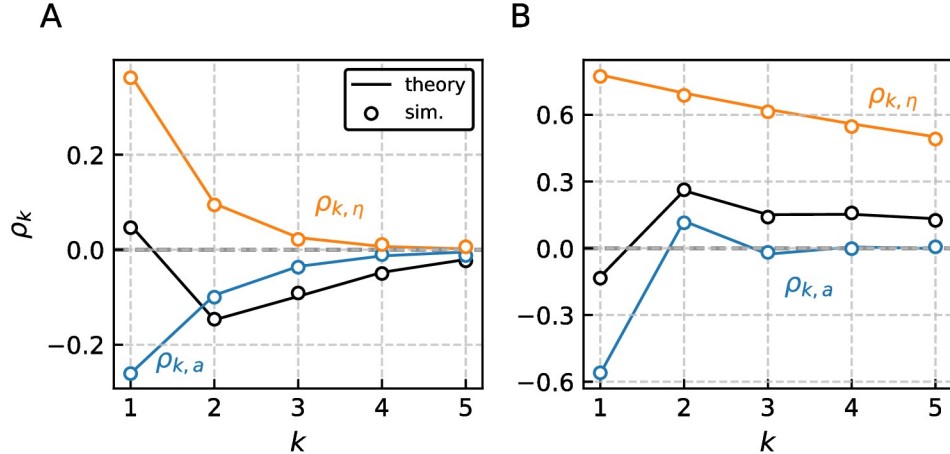

**Fig 3. General correlation coefficient $\rho_k$ of the adaptive LIF model subject to white and colored noise.** The specific SCCs $\rho_{k,a}$ and $\rho_{k,\eta}$ are obtained by considering one correlation inducing process at a time, i.e. adaptation and white noise or colored and white noise, respectively. Two qualitatively different cases are displayed distinguished by i) the base pattern exhibited by $\rho_{k,a}$ that is exponentially decaying in A and oscillatory in B and ii) the sign of the first ($k = 1$) and every subsequent ($k \geq 1$) SCC. For example consider A where $\rho_1 > 0$ and $\rho_k < 0$ for $k > 0$. The inverse case is shown in B, i.e. negative correlations at lag 1 and positive ISI correlation for every subsequent lag. Such patterns have been reported in cats peripheral auditory fibers and the weakly electric fish electroreceptors [50, 51] and can not be explained by adaptation or colored noise alone. Note that the SCC of the full model is not bound by the specific SCCs. Parameters (A, B): $\gamma = 1$, $\mu = (5, 20)$, $\tau_a = (2, 1)$, $\Delta = (2, 10)$, $\tau_\eta = (0.5, 5)$, $\sigma^2 = 2 \cdot 10^{-2}$, $D = 10^{-3}$.

## Adaptive leaky integrate-and-fire model with colored noise

For one-dimensional IF models, i.e. $f_0(v, \mathbf{w}) = f(v)$ the PRC can be calculated analytically by means of the adjoint method Eq (17)

$$Z(\tau) = Z(T^*)\exp\left[\int_\tau^{T^*} d\tau' f'(v(\tau'))\right], \tag{7}$$

where the prime denotes the derivative with respect to $v$ and $Z(T^*) = [f(v_T) + \mu - a^* + \Delta/\tau_a)]^{-1}$ is the inverse velocity $\dot{v}(T^*)^{-1}$ of the deterministic system at the threshold, see Eq (20). The term $a^* - \Delta/\tau_a$ corresponds to $a(T^*)$ right *before* the spike (recall that $a^*$ is the deterministic peak adaptation value right *after* the spike).

Since the neuron is required to fire in the absence of noise this velocity is positive and so is the PRC. Put differently, for every one-dimensional IF model, a positive kick in the voltage variable will always advance the phase as it brings the neuron model closer to the threshold.

For the adapting leaky IF model ($f_0(v) = -\gamma v$, $N = 0$) in particular, the PRC reads [39]

$$Z(t) = \frac{\exp[\gamma(t - T^*)]}{\mu - \gamma v_T - a^* + \Delta/\tau_a}. \tag{8}$$

As discussed in the previous section the specific SCC $\rho_{k,a}$ is a geometric sequence with oscillatory or exponential base pattern, distinguished by $v < 0$ and $v > 0$, respectively. Its prefactor (using $|\alpha v| < 1$ and $0 < \alpha < 1$)

$$-E(1 - v) \quad \text{with} \quad E = \frac{\alpha(1 - \alpha^2 v)}{1 + \alpha^2 - 2\alpha^2 v} > 0$$

depends specifically on the PRC and is always negative for positive PRCs. This is so because according to Eq (4d) for positive PRCs we find $v < 1$. The second term $\rho_{k,\eta}$ decays

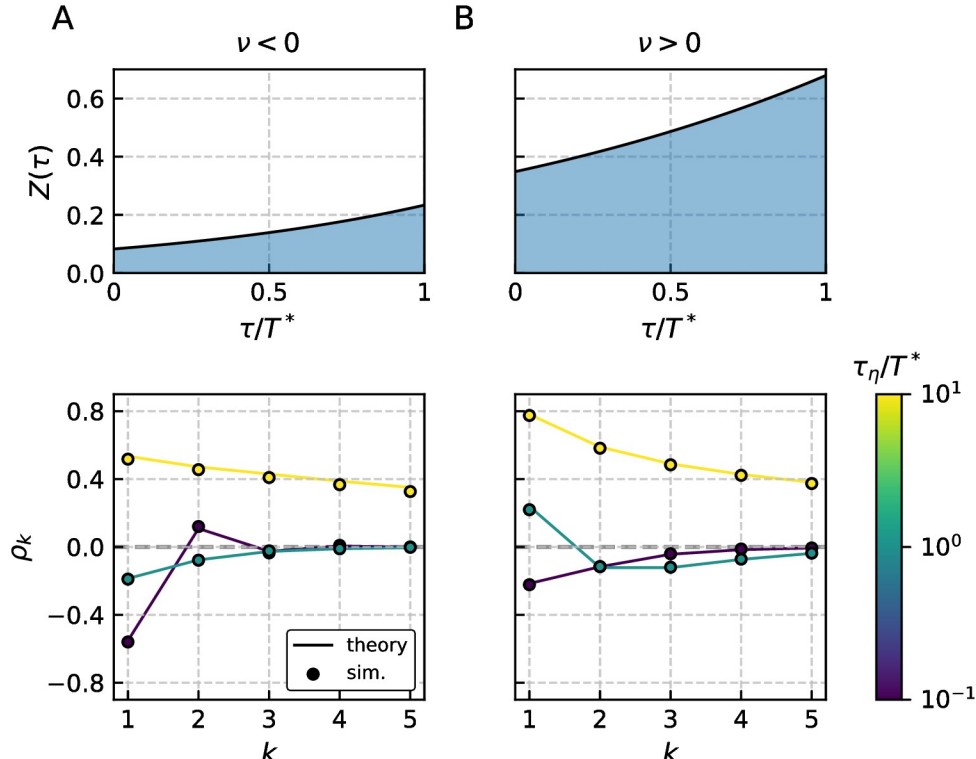

**Fig 4. Pattern of interval correlations for the adaptive LIF model.** The PRC $Z(\tau)$ and SCC $\rho_k$ for two different cases that are strong $\nu < 0$ A and weak adaptation $0 < \nu < 1$ B are shown. In both cases the colored noise correlation time $\tau_\eta$ is gradually increased. For small correlation times the SCC is governed by the adaption as the colored noise becomes essentially white (dark line and circles). In the other limit of long correlation times the SCCs are positive and governed by the colored noise (light line and circles). For intermediate time scales the SCCs are determined by both processes equally as shown in Fig 3. Parameters (A, B): $\gamma = 1$, $\mu = (20, 5)$, $\tau_a = 2$, $\Delta = (20, 2)$, $\sigma^2 = 0.1$, $D = 0$ and resulting $T^* = (0.67, 1.04)$.

exponentially with lag $k$ and possesses a non-negative prefactor because of the positive PRC of the considered model (this becomes evident in Eq (32)). To summarize, in a type I neuron model with non-negative PRC correlated noise leads to positive interval correlations with time constant equal to the correlation time of the noise. On the contrary, spike-triggered adaptation evokes a negative correlation between adjacent intervals ($k = 1$) followed by an exponential decay in amplitude at higher lags ($k > 1$) that can be either monotonic or oscillatory.

The relation between specific and general SCCs for the LIF model was already discussed in the preceding section; cf. Fig 3. In Fig 4 we inspect how the general correlation coefficient depends on the correlation time of the colored noise $\tau_\eta$, here given in multiples of the unperturbed period $T^*$. We distinguish the two possible cases of strong Fig 4A and weak adaptation Fig 4B in terms of the parameter $\nu$ that we can recast into $\nu = (f(v_R) + \mu - a^*)Z(0)$ (derivation similar to [38], Appendix 4.1.2). Strong and weak adaptation are related to the sign of the parameter $\nu$ that appears for the noiseless system in the temporal derivative of the voltage at the reset $\dot{v} = \nu/Z(0)$. For weak adaptation ($\nu > 0$) the voltage after reset runs on average towards the threshold. On the contrary, for strong adaptation ($\nu < 0$) the peak adaptation value $a^*$ is so high that the voltage after being reset to $v_R$ drops on average even further towards more hyperpolarized values.

For $\tau_\eta \ll T^*$ the noise becomes effectively white and consequently correlations are introduced solely by adaptation, i.e. the general SCC is reduced to $\rho_{k,a}$. In the other limit of long

correlated noise with $\tau_\eta \gg T^*$, the general SCC does coincide with $\rho_{k,\eta}$ if there is no white noise present ($D = 0$). This implies that in the limit of long correlated noise the origin of positive ISI correlations (the colored noise) wins against the origin of negative ISI correlations (the adaptation), and consequently, $\rho_k > 0$. Why does the colored noise dominate in this limit? A long-range correlated noise $\eta(t)$ can be regarded as a constant modulation of the input $\mu$ over many ISIs leading to similar deviations in adjacent intervals from the mean ISI. The adaptation acting on a finite scale $\tau_a$ will reduce these deviations in adjacent intervals, but can not change their common sign.

For intermediate values $\tau_\eta \approx T^*$ the SCCs can be governed by one process for small lags and the other for higher lags which is the case shown in Fig 4B. The first SCC is governed by positively correlated noise while the remaining SCCs are negative due to adaptation.

## Adaptive generalized integrate-and-fire model with colored noise

In the two-dimensional case, as for instance for the generalized IF (GIF) model [53, 54]

$$f_0(v, w) = -\gamma v - \beta_w w \tag{9a}$$

$$f_1(v, w) = (v - w)/\tau_w \tag{9b}$$

the PRC can be partially negative and resembles type II resetting. Positive kicks applied to the voltage at appropriate time instances can thus prolong the ISI. The PRC can be calculated analytically [39]:

$$Z(t) = \frac{e^{\frac{\lambda}{2}(t-T^*)}\left[\cos(\Omega(t - T^*)) - \frac{1-\tau_w\gamma}{2\tau_w\Omega}\sin(\Omega(t - T^*))\right]}{\mu - \gamma v_T - \beta_w w_0(T^*) - a^* + \Delta/\tau_a} \tag{10}$$

where $\lambda = \gamma + 1/\tau_w$, $\Omega = \sqrt{\frac{\beta_w+\gamma}{\tau_w} - \frac{\lambda^2}{4}}$ and $w_0(T^*)$ is the deterministic value of the auxiliary variable $w$ at the threshold. Having a second variable not only changes the PRC qualitatively but will also affects the deterministic period $T^*$ and, consequently, the parameters $\alpha$ and $\beta$. We would like to emphasize that the GIF model includes the LIF model as a limit case. For this reason we expect that all patterns shown by the LIF model can also be realized by the GIF model.

The observed patterns for $v < 0$ and $0 < v < 1$, shown in Fig 5A and 5B, can also be realized by the LIF model and have been discussed in the preceding section (details depend on the specific parameter and model, though).

As a consequence of the partially negative PRC, the parameter $v$ can exceed one as seen from Eq (4d) and introduce positive correlations even if the driving noise is only shortly correlated. This is seen in Fig 5C where the correlation coefficients for short correlation times are positive. We recall that this case cannot be realized by an LIF model with adaptation and thus represents a novel feature of the GIF model. The involved dependence of $\rho_1$ on the correlation time $\tau_\eta$ is presented in a different way in Fig 6A and contrasted with a similar case in the absence of adaptation in Fig 6B.

In order to understand the case $v > 1$ shown in Fig 5C, first consider the effect of the adaptation separately from the colored noise. A shortened reference interval evokes a positive deviation of the peak adaptation value, $\delta a_i = a_i - a^*$ affecting the next interval. If the corresponding inhibitory (negative) current $[-\delta a_i \exp(-\tau/\tau_a)$, see Methods], acts mainly at the beginning of the next interval where the PRC is negative as well (for instance with $\tau_a \approx T^*/2$ as in Fig 5C), it has a *shortening* effect on the subsequent interval. First and second interval are both shorter than the mean, implying a positive correlation; a similar line of arguments applies for a reference interval longer than the mean ISI. Now consider the additional effect of the correlated

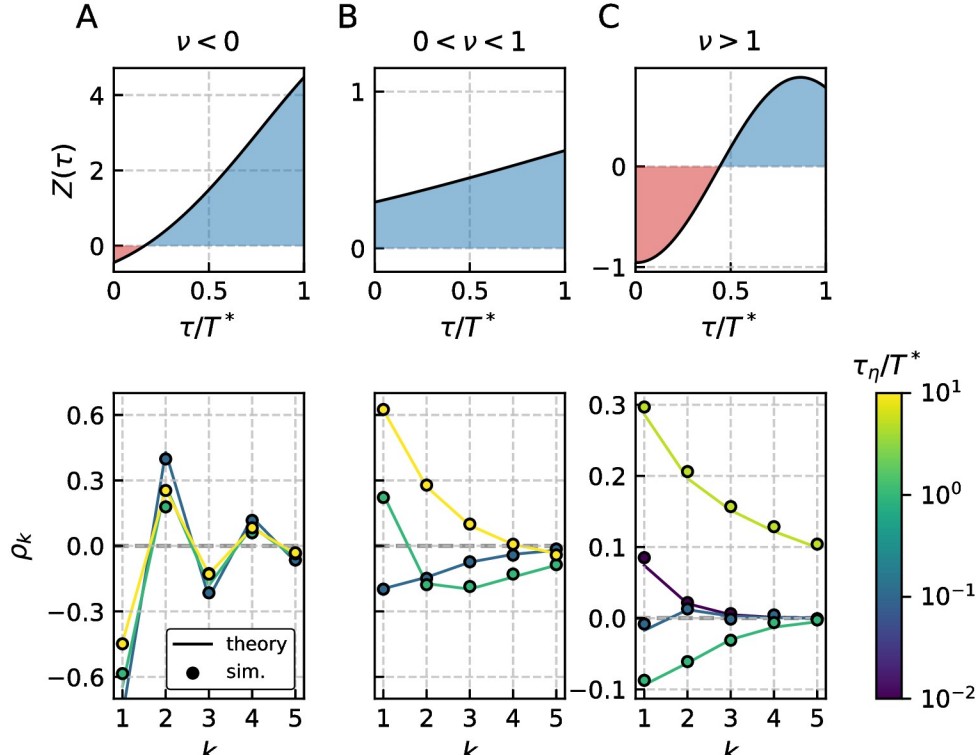

**Fig 5. Pattern of interval correlations for the adaptive GIF model.** The PRC $Z(\tau)$ and SCC $\rho_k$ for three different cases that corresponds to $\nu < 0$ A, $0 < \nu < 1$ B and $\nu > 1$ C are shown. The first two cases A, B resemble the previously discussed cases of the adaptive LIF model, see Fig 4. For the third case C both adaptation and correlated noise can have counter intuitive effects on the SCC if they act mainly on the proportion of the ISI where the PRC is negative. This can be ensured by appropriate choice of the time scales, here $\tau_a \approx T^*/2$ and $\tau_\eta \approx T^*/2$. Thus the adaptation can give rise to positive interval correlations and additional colored noise with varying correlation time initially decreases the SCCs for intermediate $\tau_\eta$ and eventually leads to enhanced positive correlations for large $\tau_\eta$. Parameters (A, B, C): $\gamma = (1, 1, -1)$, $\mu = (10, 20, 1)$, $\beta_w = (3, 1.5, 5)$, $\tau_w = (1.5, 1.5, 1.1)$, $\tau_a = (10, 10, 1)$, $\Delta = (10, 10, 2.3)$, $\sigma^2 = 10^{-3}$, $D = 0$ and resulting $T^* = (1.24, 0.57, 1.91)$.

noise. First, short-range correlated noise ($\tau_\eta/T^* = 10^{-2}$) is essentially white and does not introduce correlations. Therefore the SCC is governed by adaptation and remains positive by the mechanism discussed above. Secondly, consider larger correlation times still shorter than the proportion of the ISI for which the PRC is negative, e.g. $\tau_\eta/T^* = 10^{-1}$. Values of $\eta(t)$ that are preserved beyond a spike will then have opposite effects on the two intervals separated by the spike, inducing an anti-correlation of these intervals. As a consequence the overall SCC decreases initially with increasing $\tau_\eta$. Finally, for correlation times equal or longer than the mean ISI, $\tau_\eta \geq T^*$, the particular shape of the PRC becomes irrelevant and the positive correlations of the colored noise translate into positive correlations of the ISIs. The minimal correlation at intermediate values of the colored noise correlation time is demonstrated in Fig 6A. The asserted anti-correlation induced by a colored noise of intermediate correlation time is explicitly demonstrated in Fig 6B. Here we consider the GIF model without adaptation at parameters that ensure a negative PRC at short times. Clearly, the SCC is negative for intermediate values of the correlation time. This is an interesting result in its own right: A low-pass filtered noise can evoke negative correlations in a resonator model. Besides the combinations of white noise and spike-triggered adaptation [16, 55, 56], white noise and short term depression

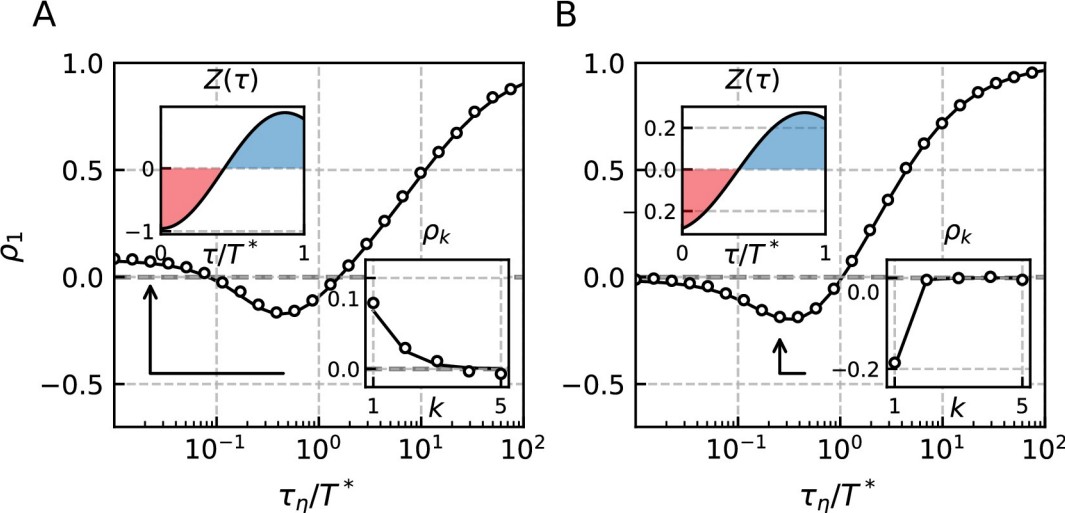

**Fig 6. First serial correlation coefficient of the GIF model with respect to the correlated noise time constant $\tau_\eta$.** Panel A shows the SCC for an adaptive GIF with parameters similar to those in Fig 5C. In panel B we consider a GIF model without adaptation and parameters chosen so that the PRCs in A and B qualitatively agree. The SCC at lag $k = 1$ can exhibit non-monotonic behavior with respect to the time constant $\tau_\eta$ due to the partially negative PRC. The PRC is shown in the upper left inset and is in both cases found to be negative until $\tau \approx T^*/2$. If the correlation time matches this proportion of the ISI the SCC is significantly decreased compared to the case of short correlation times. For $\tau_\eta \gg T^*$ adjacent ISIs are positively correlated as they are similarly affected by the slow varying noise. Parameters (A, B): $\gamma = -1$, $\mu = 1$, $\beta_w = 5$, $\tau_w = (1.1, 1.1)$, $w_R = (0, 1)$, $\tau_a = (1, 0)$, $\Delta = (2.3, 0)$, $\sigma^2 = 10^{-3}$, $D = 0$ and resulting $T^* = (1.91, 1.76)$.

[47], and network noise from neurons firing more regular than a Poisson process [47], this is yet another independent mechanism for the generation of negative ISI correlations.

## Leaky integrate-and-fire model with adaptation-channel noise

Adaptation and colored noise in our model Eq (2) can also be regarded as an idealized description of the current flowing through a population of stochastically opening and closing ion channels with adaptation-mediating voltage-dependent gating kinetics [33, 34]. A paradigmatic example is the $Ca^{2+}$-dependent $K^+$ current [55, 57] but several other candidates for such spike-triggered adaptation currents are known (see [35]). Whatever the type of channel is, the current through a *single* channel is highly stochastic and this remains true also for the summed current through a *finite* population of channels—this is what is commonly referred to as channel noise. As was demonstrated in [33], the total current through a finite population of adaptation channels can be split up into a deterministic part (equivalent to the adaptation dynamics Eq (2c)) and a stochastic part that corresponds to an Ornstein-Uhlenbeck process (our Eq (2d)).

In this interpretation of the model, the summed current $a(t) + \eta(t)$ stems from one source, i.e. from the population of adaptation channels and we regard the sum as adaptation-channel noise (the white noise may be regarded as resulting from faster, e.g. $Na^+$ channels). Crucially, because noise and adaptation have a common origin, the previously independent time constants $\tau_a$ and $\tau_\eta$ have to be set equal. We will refer to the common time constant as $\tau_c := \tau_a = \tau_\eta$. Note that consequently we have $\alpha = \beta$, parameters which where defined in Eq (4d). We emphasize that we will not consider explicit channel models here but refer the interested reader to [33, 34].

First note that our expression for the general SCC Eq (3) simplifies considerably if $\tau_a = \tau_\eta$ since in this case the second term $(B/C)\rho_{k,\eta}$ drops out. This is so because with $\alpha = \beta$ the

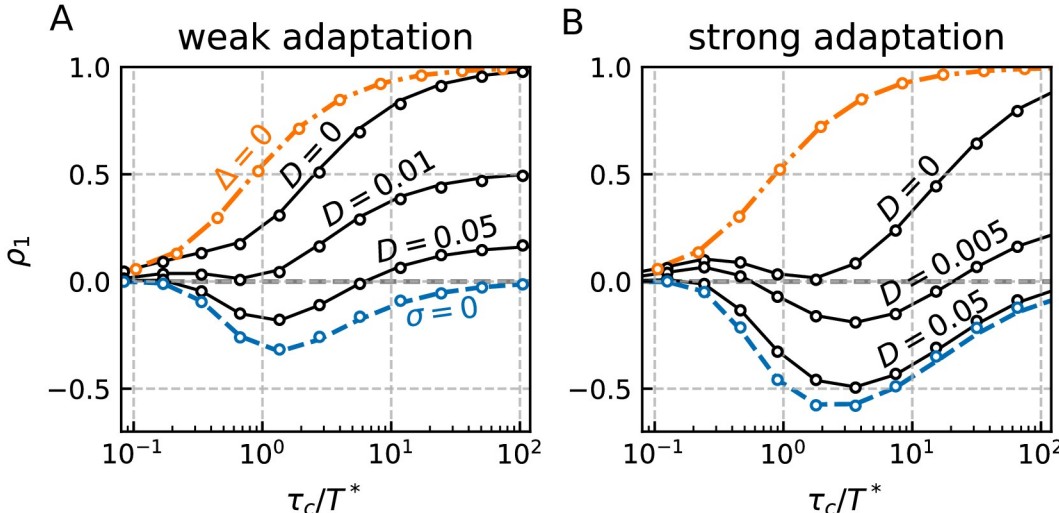

**Fig 7. Serial correlation coefficients for the LIF with adaptation-channel noise.** The SCC of adjacent intervals $\rho_1$ (black lines and dots) for our model Eq (2) with identical time constants $\tau_\eta = \tau_a = \tau_c$ shows non-monotonic behavior with a minimum as a function of $\tau_c$ given that the white noise intensity $D$ is sufficiently large. This is so because the kick amplitude scales with $\tau_c^{-1}$. The panels A and B correspond to weak and strong adaptation, respectively. The limiting cases are shown in orange (no adaptation or white noise, $\Delta = 0$, $D = 0$) and blue (no colored noise, $\sigma = 0$, $D = 0.1$). However, only the limit case of vanishing colored noise can be attained by the full model through varying the white noise intensity. Parameters: (A, B) $\gamma = 1$ $\mu$ = (5, 20), $\Delta = (2, 20)$, $\sigma^2 = 0.1$. Note that in contrast to the Figs 4–6, the deterministic period $T^*$ depends for some of the curves on $\tau_c$. For the black and blue curves the deterministic period $T^* \in [0.56, 0.67]$ in A and $T^* \in [0.74, 1.08]$ in B and increases in both panels with $\tau_c$. For the orange curve $T^* = 0.22$ in A and $T^* = 0.05$ in B.

prefactor $B = 0$, see Eq (4b). The possible interval correlations are thus determined by a single geometric sequence and comprise exponentially decaying or oscillatory patterns.

Quantitative agreement between simulations (circles) and theory (lines) for the first correlation coefficient is demonstrated in Fig 7 for the cases of weak (Fig 7A) and strong adaptation (Fig 7B) and varying intensities of white noise (black lines). We also compare to the limiting cases of vanishing colored fluctuations ($\sigma = 0$, blue dashed line) and vanishing adaptation ($\Delta = 0$, orange dash-dotted line).

We recover a number of results known from the literature. First of all, if the total noise in the system is dominated by the uncorrelated fluctuations ($D$ is sufficiently large), the SCC is negative and the absolute value is maximized if the time constant of the adaptation is about the mean ISI [33, 55]. Secondly, as already argued above and in line with results from [33] the correlations are always positive if $\tau_c$ is sufficiently large and the white noise is sufficiently weak, i.e. the stochasticity of the adaptation (described by the colored noise) wins against the feedback effect of the adaptation in determining the sign of the correlation coefficient. Finally, we find qualitative agreement of $\rho_1(\tau_c)$ in our model with the channel model of Ref. [33]: it exhibits a non-monotonic shape with small negative correlations for small $\tau_c$ and positive correlations for larger values of $\tau_c$ (cf. solid line for D = 0.05 in Fig 7A, with empty circles in Figure 9 of Ref. [33]).

We would like to emphasize that for the channel noise case we have explicitly taken into account additional white noise (we used $D = 0$ for the previous cases). As it becomes evident from Fig 7, this white noise can change the sign of the correlation coefficient and, more generally, the way the first SCC depends on the time constant of the channel kinetics. This illustrates how different channel fluctuations may interact to shape the serial correlation coefficient of the interspike interval.

## Adaptive leaky integrate-and-fire model with network-noise-like fluctuations

So far we have discussed neuron models that are subject to positively correlated noise as it would arise due to synaptic filtering of uncorrelated pre-synaptic spike trains or due to slow adaptation channels. Such input noise processes exhibit power spectra with *increased* power at low frequencies (or, depending on the perspective, *reduced* power at *high* frequencies). This has clear applications to neurons in the sensory periphery that often lack synaptic input (spike variability is mainly caused by channel noise) or are only subject to approximately uncorrelated feedforward spike input. However, what about the interesting case of cortical neurons as part of a recurrent network of neurons? Although our theory requires a mean-driven cell and is therefore *not generally* applicable to this situation (many neurons in the cortex operate in an excitable, i.e. fluctuation-driven, regime), we discuss now a special case in which it nevertheless can be employed.

 Most neural networks show a large heterogeneity with respect to cellular parameters as well as to the kind, number, and strength of synaptic connections, resulting in distributions of firing rates and CVs with the latter measure of variability varying between 0.2 and 1.5 (see e.g. [58]). For cells that fire rather regular and with high rates we can assume that they are effectively mean driven (with respect to the sum of intrinsic and recurrent currents); although a low value of the $C_V$ is in principle also possible for an excitable neuron through the mechanism of coherence resonance [59], this requires a close proximity to the bifurcation point and a fine tuning of the noise intensity that is unlikely to take place in the mentioned cells. The same argument can be made for certain areas in the brain, as for instance the motor cortex, where the firing variability is generally lower (see e.g. the review [60] or a more recent study on the variability in motor cortex [61]).

 We consider such a mean-driven neuron that receives input from a recurrent network in an asynchronous irregular state (the neuron could be part of this network or be subject to a feedforward input from such a network). The kind of network noise can be approximated by a Gaussian process; its temporal correlation function can attain different shapes and depends on the detailed connectivity in the network (see e.g. [62, 63]). In typical cases of low to intermediate firing rate one encounters both in *in vivo* experiments [64] and in theoretical studies [45, 62, 65] fluctuations that are referred to as *green noise*, the power spectrum of which possesses reduced power at low frequencies and is otherwise flat. Such a power spectrum is well approximated by a sum of a white noise and an Ornstein-Uhlenbeck process (see e.g. Figure 14 in [45]) if the previously independent noise sources in Eq (2a) and (2d) are chosen to be anti-correlated $\xi_\nu = -\xi_\eta$. The respective power spectrum of the random process $\zeta = \eta(t) + \sqrt{2D}\xi(t)$ is given by

$$S_{\zeta\zeta}(\omega) = 2D + \frac{2\tau_\eta\sigma^2 - 4\sqrt{D\tau_\eta\sigma^2}}{1 + \tau_\eta^2\omega^2}. \tag{11}$$

This power spectrum possesses a constant high frequency limit $\lim_{\omega\to\infty} S(\omega) = 2D$ and reduced power at low frequencies $S_{\zeta\zeta}(0) = (\sqrt{2D} - \sqrt{2\tau_\eta\sigma^2})^2 < 2D$, see Fig 8A. The turning point between those two limits is given by $\omega = 1/\tau_\eta$. Furthermore, numerical inspection of the self-consistent network spectrum in [45] revealed that the parameter $\tau_\eta$ is mainly set by the mean firing rate $r_0$ of the neurons in the recurrent network, or, equivalently, to their mean ISI $\langle T \rangle_\text{net} = 1/r_0$, and we have roughly $\tau_\eta \approx \langle T \rangle_\text{net}/\pi$. The parameters $D$ and $\sigma^2$ are determined by the number, type, and strength of synaptic connections, see [45].

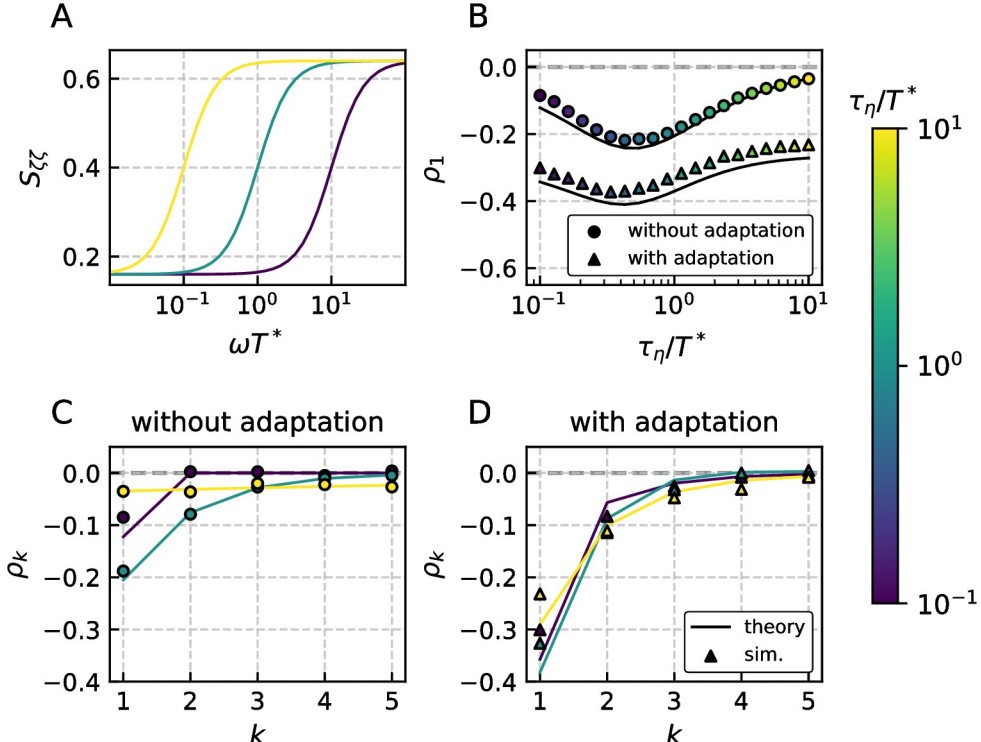

**Fig 8. Serial correlation coefficients for the LIF with network-noise-like fluctuations.** Panel A shows the input power spectrum $S_{in}$ with reduced power at low frequencies of a random process that is applied to a LIF model. The resulting SCC $\rho_1$ for no adaptation (upper line) and weak adaptation (lower line) are displayed in panel B. The coefficient $\rho_1$ has a minimum because for both long and short correlation times the noise becomes essentially white. The pattern of interval correlations are shown in C for no and D for weak adaptation. Even in the absence of an adaptation current the network-noise can generate negative ISI correlations. This is due to the lack of power at low frequency. For an adaptive LIF model negative correlations are enhanced. Parameters (C, D): $\gamma = 1$, $\mu = 5$, $\tau_a = (0, 2)$, $\Delta = (0, 2)$, $D = 0.18$, $\tau_\eta \sigma^2 = 4.5 \cdot 10^{-2}$ and resulting $T^* = (0.22, 0.67)$.

It turns out that such a power spectrum can be treated by our theory if we adjust the specific correlation coefficient $\rho_{k,\eta}$ as follows

$$\rho_{k,\eta} = \frac{\int d\omega |\tilde{Z}(\omega)|^2 S_{\zeta\zeta}(\omega) e^{-ik\omega T^*}}{\int d\omega |\tilde{Z}(\omega)|^2 S_{\zeta\zeta}(\omega)}. \tag{12}$$

This is exactly the approximation for the SCC that has been derived for purely colored-noise driven neurons in [47]. For our model with an adaptation variable, this generalization is valid if the corresponding autocorrelation function $C_{\zeta\zeta}(\tau)$ decays exponentially (an additional delta-function is also permitted), which is the case for the considered green noise.

In Fig 8 we consider a mean-driven LIF neuron with (Fig 8C) or without adaptation (Fig 8D), which is subject to network-noise-like green fluctuations with spectrum $S_{\zeta\zeta}$, see Fig 8A. This spectrum exhibits a low-frequency power suppression of $S_{\zeta\zeta}(\omega = 0)/S_{\zeta\zeta}(\omega \to \infty) = 0.25$, similar to that of the self-consistent power spectrum in Ref. [45], Fig 14. We choose the overall amplitude of the noise such that the resulting CV is in between 0.2 and 0.5, which is in the lower physiological range for cortical cells. We compare the theoretical SCC (lines) to stochastic simulations for different values of the time scale $\tau_\eta$ and find in all cases a good agreement; this becomes an excellent agreement for weaker noise, as can be expected. Remarkably, even in the absence of adaptation, a green noise evokes negative ISI correlations, see Fig 8C; a

corresponding observation has been made for another noise process with reduced low-frequency power in [47] and also the excitable (non-adapting) cells in the recurrent network in Ref. [45] exhibit a (somewhat smaller) negative correlation of $\rho_1 \approx -0.1$ [66]. With an additional adaptation current (Fig 8D), negative ISI correlations become even stronger as can be expected. Furthermore, in both cases the SCCs depend non-monotonically on $\tau_\eta$, see Fig 8B because, somewhat non-intuitive, in both limits $\tau_\eta \to 0$ and $\tau_\eta \to \infty$ the effective noise becomes white (uncorrelated) and will not cause negative correlations anymore. We note that for the network situation with a mean-driven cell with mean ISI $T^*$ firing somewhat faster than the average cell with ISI $\langle T \rangle_{\text{net}}$, a time constant $\tau_\eta = \langle T \rangle_{\text{net}}/\pi \gtrsim T^*/\pi$, e.g. a ratio of $\tau_\eta/T^* = 1$ seems to be the most relevant value. Interestingly, this is close to the value that maximizes the strength of correlations, cf. Fig 8B.

## Adaptive generalized integrate-and-fire model with both colored and white noise—Testing the range of validity

We turn to the most general case and discuss to what extend it can be expected that the SCC of a simulated stochastic spike train is well described by our perturbation theory. We choose a specific parameter set with respect to the deterministic system and the involved time scales and vary the two small parameters of our theory, i.e. the white noise intensity $D$ and variance of the colored noise $\sigma^2$. In order to inspect the range of validity, we show not only the SCC $\rho_1$ but also a popular measure of the output variability, the coefficient of variation (CV)

$$C_V = \frac{\sqrt{\langle (T_i - \langle T_i \rangle)^2 \rangle}}{\langle T_i \rangle}.$$

In a first simulation setup the variance $\sigma^2$ is fixed so that for small values of $D$ the SCC $\rho_1$ is predominantly determined by the colored noise (cf. $\rho_1 > 0$ in Fig 9A bottom for small $D$). Increasing the white noise intensity has a twofold effect. First, it boosts the output variability of the spike train (cf. growth of the CV in Fig 9A top). Secondly, with stronger white noise the adaptation becomes the dominant process in shaping the SCC ($\rho_1 < 0$) as demonstrated in Fig 9A bottom for large $D$. Put differently, the sign of the SCC can be determined by the ratio of the noise intensities $D/(\tau_\eta \sigma^2)$, which can be seen from Eq (6), the only place where the noise intensities enter in our theory.

We find quantitative agreement between theory and simulation for both the CV and SCC $\rho_1$ up to $C_V = 0.3$. Qualitative agreement is maintained over the whole tested range of noise intensities $D$; even for a relatively high CV, e.g. $C_V \approx 0.7$, the dependence of the SCC on the lag $k$ is well described by our theory (see inset Fig 9A).

In a second setup we fix the white noise intensity $D$ and test how varying the colored-noise variance affects the agreement between simulation and theory.

Here we find quantitative confirmation of our theory up to $C_V = 0.15$. Again, theory and simulations agree qualitatively over the whole range tested as demonstrated in the inset of Fig 9B. Specifically, focusing on the SCC's dependence on the lag $k$ the theory reproduces the change in sign between $\rho_1$ and $\rho_2$, the minimum at $k = 3$ and the subsequent decay.

## Traub-Miles model with an M current and both colored and white noise

Finally, we demonstrate that our theory can be applied beyond the integrate-and-fire framework to a Hodgkin-Huxley-like conductance-based neuron. Specifically, we use the Traub-Miles model endowed with a slow adaptation-like potassium current as considered by

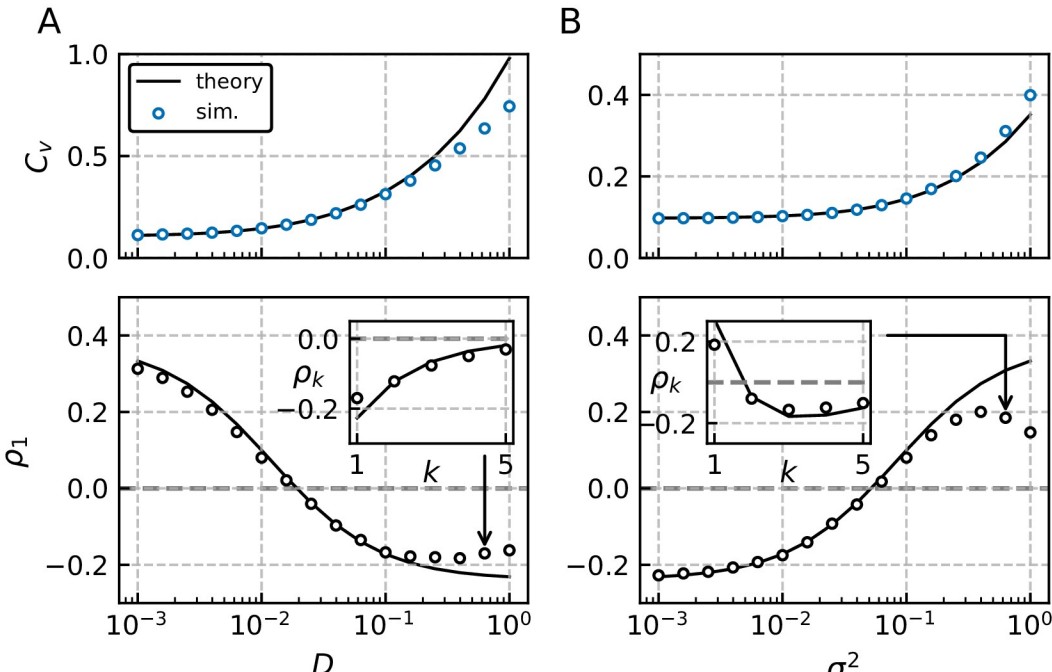

**Fig 9. Range of validity and effect of varying noise strengths.** Coefficient of variation $C_V$ (top) and SCC of adjacent intervals $\rho_1$ (bottom) for the GIF model with weak adaptation. We test to which extent simulation results are well described by our weak-noise theory with respect to the two small parameters ($D$, $\sigma^2$). In A the variance of the colored noise is fixed ($\sigma^2 = 0.1$) and the white noise intensity $D$ is varied; in B we fix $D = 0.01$ and vary $\sigma^2$. We find quantitative agreement for $C_V < 0.3$ in A and $C_V < 0.15$ in B. In both cases qualitative agreement in terms of the shape of the SCCs (see insets) is found over the whole range of $D$ and $\sigma^2$, respectively. Parameters: $\gamma = 1$, $\mu = 20$, $\beta_w = 1.5$, $\tau_w = 1.5$, $\tau_a = 10$, $\Delta = 10$, $\tau_\eta = 1$ and resulting $T^* = 0.57$.

Ermentrout [67] and drive it with both colored and white (notation as above)

$$C\frac{dV}{dt} = -I_{\text{ion}} - I_{\text{adap}} + I + \eta + \sqrt{2D}\xi_V(t), \tag{13a}$$

$$\tau_\eta \frac{d\eta}{dt} = -\eta + \sqrt{2\tau_\eta \sigma^2}\xi_\eta(t). \tag{13b}$$

Here $I$ is a constant current and $I_{\text{ion}}$ comprises the fast sodium, leak and potassium currents, given in terms of the reversal potentials $E_y$, the maximum conductances $g_y$ and the gating variables $h$, $n$ and $m$

$$I_{\text{ion}} = g_{\text{Na}}hm^3(V - E_{\text{Na}}) + g_{\text{L}}(V - E_{\text{L}}) + g_{\text{K}}n^4(V - E_{\text{K}}), \tag{14a}$$

$$\frac{dx}{dt} = a_x(V)(1 - x) - b_x(V)x, \quad \text{with} \quad x = h, m, n. \tag{14b}$$

**Table 1. Simulation parameters for the Traub-Miles model.**

| Parameter | Value | Parameter | Value | Parameter | Value |
|---|---|---|---|---|---|
| $I$ $[\mu A/cm^2]$ | 5 | $g_{Na}$ $[mS/cm^2]$ | 100 | $a_h(V)$ | $0.128\exp\left(-\frac{V+50}{18}\right)$ |
| $D$ $[(\mu A/cm)^2 ms]$ | 0.1 | $g_L$ $[mS/cm^2]$ | 0.1 | $a_m(V)$ | $0.32\frac{V+54}{1-\exp\left(-\frac{V+54}{4}\right)}$ |
| $\tau_\eta$ $[ms]$ | 10 | $g_K$ $[mS/cm^2]$ | 80 | $a_n(V)$ | $0.032\frac{V+52}{1-\exp\left(-\frac{V+52}{5}\right)}$ |
| $\sigma^2$ $[(\mu A/cm)^2]$ | 0.1 | $E_{Na}$ $[mV]$ | 50 | $b_h(V)$ | $\frac{4}{1+\exp\left(-\frac{V+27}{5}\right)}$ |
| $\tau_z$ $[ms]$ | 100 | $E_L$ $[mV]$ | -67 | $b_m(V)$ | $0.28\frac{V+27}{\exp\left(\frac{V+27}{5}\right)-1}$ |
| $\bar{g}$ $[mS/cm^2]$ | 5 | $E_K$ $[mV]$ | -100 | $b_n(V)$ | $0.5\exp\left(-\frac{V+57}{40}\right)$ |

spike-frequency adaptations is mediated by a slow potassium current

$$I_{adap} = \bar{g}z(V - E_K),\tag{15a}$$

$$\tau_z\frac{dz}{dt} = h(V) - z,\quad \text{with}\quad h(V) = \frac{1}{1+\exp\left(-\frac{V+20}{5}\right)}.\tag{15b}$$

For $a_x(V)$, $b_x(V)$, $h(V)$ and the parameter values, see Table 1.

The transient behavior of $V(t)$ and $z(t)$ in response to a current step, see Fig 10A, displays spike-frequency adaptation (note the increase in the interspike intervals in the course of time). The noisy trajectory of the full system is close to the deterministic limit cycle for a sustained constant input, Fig 10B. Using the deterministic model ($D = 0$, $\sigma = 0$) and small short pulses, we can determine the PRC of the system according to Eq (16) numerically, see Fig 11A.

In addition to the PRC our theory requires the knowledge of the deterministic interspike interval as well as the peak-value and the time-scale of the adaptation current. The latter time-scale is readily identified as the time-constant of the gating variable $z$, $\tau_a = \tau_z$. The former parameters can be determined numerically from simulations of the deterministic model; we

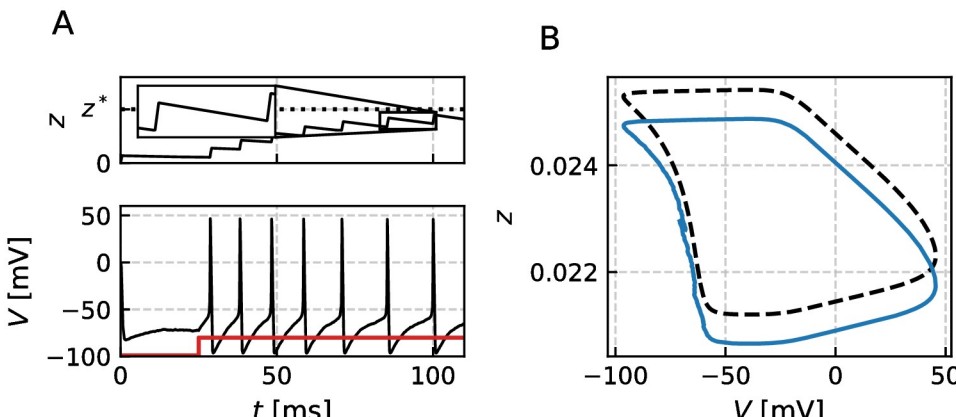

**Fig 10. Traub-Miles model with slow M current.** Panel A shows the membrane potential $V(t)$ and the adaptation's gating variable $z(t)$. At $t = 25$ms a constant current $I = 5\mu A/cm^2$ (red line panel) is applied so that the model undergoes a transition from the excitable to the tonic firing regime. Due to the slow build-up of the adaptation current the model shows a transient behavior where the firing-rate decreases until $z(t)$ has reached its stationary value (doted line). The inset shows that $z$ has two different phases, one during which $z$ rapidly increases and another where $z$ slowly decays. Panel B shows the deterministic limit cycle (dashed line) together with a noise trajectory of the tonically firing model with $T^* = 18.9$ms. Parameters are as given in Table 1.

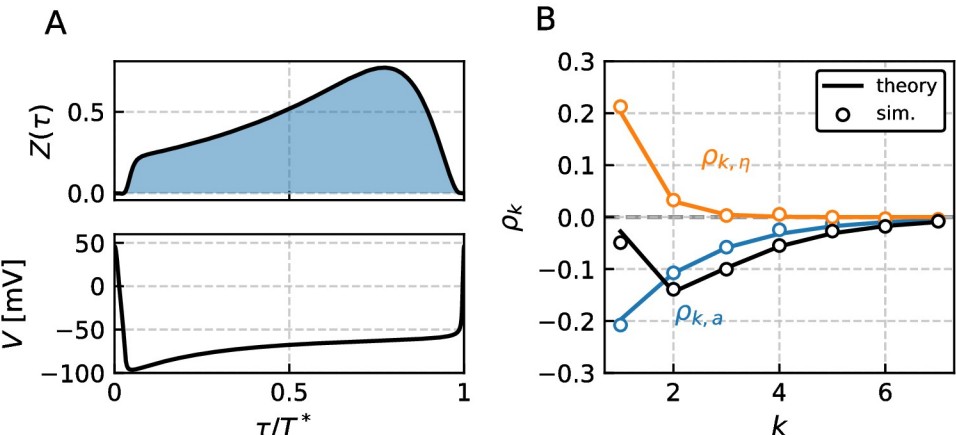

**Fig 11. Serial correlation coefficients for the Traub-Miles model with slow M current.** Panel A shows the Phase-response curve $Z(\tau)$ as well as the deterministic time-course of the membrane potential $V(t)$ from which the PRC was obtained. Note that the PRC is always positive wherefore we expect to find SCCs that are qualitatively similar to those obtained from a LIF model. Panel B shows a comparison between numerically obtained and theoretically calculated SCCs. As in Fig 3 we compare specific and general SCCs $\rho_{k,a}$, $\rho_{k,\eta}$ and $\rho_k$ and find the same patterns as for a LIF model with likewise purely positive PRC.

find $T^* = 18.9$ms and $a^* = I^*_{\mathrm{adap}}/C = 4.2$mV/ms, which correspond for the chosen parameters to a slow and weak adaptation.

Consistent with our discussion surrounding Fig 3, we inspect the SCC in three different cases: i) with adaptation and white noise, $\rho_{k,a}$ ii) with colored and white noise but no adaptation, $\rho_{k,\eta}$ and iii) with both colored and white noise and adaptation present, $\rho_k$.

The resulting SCCs, see Fig 11B, show very good agreement between simulations and theory and illustrates that our theory is applicable to conductance-based models. A comparison with Fig 3A demonstrates that the type of the PRC is indeed the essential response characteristics that determines the pattern of interspike-interval correlations.

## Discussion and conclusion

Neurons often show both stochasticity as well as slow time scales in their spiking process, features that are due to intrinsic and external noise sources ´ [4, 13] and due to adaptation currents [35], respectively. The two essential characteristics of neural firing, randomness and adaptation, do not only become apparent in the spontaneous activity of nerve cells but also influence strongly their signal transmission properties (when stimulated with time-dependent signals) [67–78] and their synchronization with other cells (when considered in large recurrent networks) [79–82]. It is thus an important goal in neuroscience to theoretically understand spiking models of the nerve cell that incorporate these features.

Multidimensional stochastic integrate-and-fire models endowed with adaptation currents and intrinsic noise are simplified yet biophysically minimalistic descriptions that incorporate both stochasticity and spike-frequency adaptation. It has been shown that they can mimic the response of pyramidal cells to complex stimuli to an astonishing degree of accuracy [5, 83–85]. Thus, not surprisingly, many theoretical efforts have been devoted to this model class, aiming to calculate the firing rate, the stationary voltage distribution, or the spike-train power spectrum. The most general theory of these statistics uses the framework of a multidimensional Fokker-Planck equation [45, 86, 87] that, however, permits in most cases only numerical solutions that do not permit simple conclusions on how certain parameters shape the SCC.

A striking effect of both adaptation currents and realistic (i.e. temporally correlated) noise sources is that the interspike intervals will be not independent of each other anymore. It is easy to understand that, for instance, a slow noise (resulting from low-pass filtered synaptic noise or from a channel population with slow kinetics) will correlate adjacent intervals. Adaptation combined with fast fluctuations, lead on the contrary to negative correlations, i.e. a short ISI is on average followed by a longer one. Often, the spike generation is more complicated, as in a resonator model [39, 53]; the noise may have fast and slow components or its power may be even concentrated in a narrow frequency band [88]; slow noise and slow adaptation may result from the same source (stochastic adaptation ion channels [33]). In all these more complicated cases, we need a theory in order to predict and interpret the sign and, more generally, the patterns of interval correlations. Special cases have been attacked with different methods, assuming a slow noise [48, 89], a weak noise [38, 39, 47, 49, 88], weak adaptation [37, 90], or a discrete Markov-state description [91, 92]. The most important problem of ISI correlations evoked by a *combination of both adaptation and colored noise* has not been addressed yet.

Here we made an important step towards a general theory of interspike-interval correlations in stochastic neurons. We derived a general formula for the serial correlation coefficient of a multidimensional IF model subject to both a spike-triggered adaptation currents and correlated noise. Two important assumptions of our theory are that i) the neuron is (if noise is switched off) in a tonically firing regime; ii) the stimulating noise is weak. We tested this formula in several situations corresponding to special cases of our general multidimensional integrate-and-fire model and we applied the theory even to a conductance-based model with adaptation. In all cases we found an excellent agreement with the theory, demonstrating that especially the second assumption is not a strong limitation. We will return to the limitations of our model below.

Our theory draws heavily on previous approaches that used the phase-response curve to study stochastic neurons [39, 47, 82]. Generalizing these methods, we arrived at a qualitatively novel result: the serial correlation coefficient as a function of the lag between the two intervals is not limited to a single geometric sequence but, if both spike-triggered adaptation and low-pass filtered noise are present, can be expressed by a sum of two geometric sequence, one corresponding to the adaptation part and one to the colored noise. Because of the nonlinear feedback nature of the adaptation and because the SCC is the ratio of two statistics (covariance and variance of intervals), this is a highly nontrivial but very useful finding. The structure of this new solution allows to explain serial correlations that have been observed experimentally but could not be explained so far theoretically. Note that this does not concern the case of a temporally structured input noise. For instance, if a neuron is driven by a narrow-band noise with power in a preferred frequency band, or a power-law noise with power distributed over a broad frequency band, the SCC adopts some of this input's temporal structure and can be more complicated [47, 88].

Correlations in the interval sequence of a spiking neuron are certainly interesting on their own: we see a complex biophysical system far from thermodynamic equilibrium that generates pulse sequences with rich statistics, very different in nature from the text book example of a Poisson process or other renewal processes that would not show any interval correlations at all. The nonlinear model with colored noise and feedback poses an important problem for theoreticians in the field of computational neuroscience. Still one may wonder why we should invest so much effort in the (approximate) calculation of this particular second-order statistics. However, besides the basic understanding of spontaneous neural firing, there are at least three more reasons to explore ISI correlations, outlined below in more detail.

First of all, having an analytical expression for the ISI correlations in terms of biophysical parameters, may allow us to extract some of these parameters from measured correlation

coefficients. That it is principally possible to extract otherwise inaccessible parameters from experimentally measured ISI correlation coefficients, has been demonstrated in Ref. [88] for electrosensory cells subject to narrowband noise. For the more involved case of an adapting neuron, it might be recommendable to use not only ISI correlations but also other statistics (spike-train power spectra, for instance) to extract model parameters. Since the correlation coefficient depends nontrivially also on simple stimulation parameters, injecting a (temporally constant) current, varying its amplitude, and comparing the resulting variation in the SCC with the theoretical formula might provide another way to access parameters.

Secondly, interval correlations in the spontaneous activity can have a strong impact on the transmission of signals. This is clearly seen in the power spectrum of the spike train, the low-frequency limit of which is directly related to the sum over the ISI correlation coefficients [93]

$$\lim_{f \to 0} S(f) = r_0 C_V^2 \left( 1 + 2\sum_{j=1}^{\infty} \rho_j \right).$$

This spectrum of the spontaneous activity plays the role of a background spectrum if a stimulus is present. Having a reduced background spectrum at low frequencies (due to negative ISI correlations) may enhance the signal-to-noise ratio in this frequency band (but also diminish it in other bands) [24, 25]; positive correlations, on the contrary, may diminish the mutual information about a broadband stimulus [94]. A similar argument for the beneficial effect of negative ISI correlations can be made for a signal detection task [15] by means of the asymptotic Fano factor $F(T) = \langle (N(T) - \langle N(T) \rangle)^2 / \langle N(T) \rangle$ (where $N(T)$ is the spike count in the time window $[0, T]$), that is simply related to the low-frequency limit of the power spectrum stated above by $\lim_{T \to \infty} F(T) = S(0)/r_0$. For the transmission of time-dependent signals, interval correlations are a way to control the noise spectrum and thus to shape the information transmission in a frequency-dependent manner, contributing to what is known as information filtering [26]. In all these cases, our theory describes how exactly the cellular dynamics (PRC), the adaptation (strength and time scale) and the noise (strength and correlation time) affect the serial correlations and by them the signal transmission properties of the respective neuron.

Thirdly, we may consider ISI correlations for neurons connected in the large recurrent networks of the brain. In networks correlations can be evoked by the slow or oscillatory (narrowband) noise emerging from the nonlinear interactions of neurons [27, 30–32, 95], by deviations of presynaptic spike statistics from Poisson statistics [47, 62], but also by synaptic short-term plasticity [47]. As outlined above for the case of a single noisy neuron, such ISI correlations can likewise affect the signal transmission properties of whole populations [27]. Moreover, the specific strength and pattern of ISI correlations may be informative about properties of the network (again in combination with other statistics). Although our theory cannot be readily applied to cortical neurons (many cells are excitable and the driving fluctuations are strong), it can be used in certain special situations, as we have demonstrated here.

Turning back to the theoretical challenges and achievements of our paper, we would like to finally discuss its limitations and possible directions of future research to go beyond the results achieved here. Although our analytical approach applies to a broad class of models and is not limited with respect to the time scales of adaptation or colored noise, it is restricted to i) neurons in the tonically firing regime and ii) weak input noise. We had to exclude cases in which neurons already in the deterministic case fire spike trains with complex patterns [96]. More restrictive from our point of view is that we had to exclude excitable neurons and neurons that are subject to strong stochastic input such as emerging from synaptic background noise in recurrent neural networks. Of course, these cases are of particular importance for the stochastic dynamics of cortical cells and they can be addressed in the Fokker-Planck framework as has

been demonstrated for important spike-train statistics in full generality in [45]. Cases in which strong noise impinges on a mean-driven neuron may be also addressed by taking into account amplitude dynamics [97, 98] and higher order phase-response functions.

However, the PRC techniques used in our work might turn out to be applicable also in these cases based on generalizations of the notion of phase to the situation of strongly stochastic and excitable neurons. Indeed, two attempts have been made over the last couple of years to generalize the phase concept for deterministic systems to stochastic oscillators: the phase can be either defined in terms of the mean-first-passage time [99–101] (see in particular the recent analytical approach to the problem in Ref. [102]) or in terms of the asymptotic evolution of the probability density [103] (see specifically the analytically tractable case of a multidimensional Ornstein-Uhlenbeck process addressed in Ref. [104]). It is yet neither clear which of the two phases is more appropriate for a given system [105, 106] nor how to generalize the concept of a phase-response curve to this case. The success of the PRC method in the limit of weakly perturbed deterministic systems demonstrated here and in previous studies [39, 47, 82] should be a strong incentive to pursue this line of research.

## Methods

### Phase response curve

The phase-response curve (PRC) provides a method to calculate the phase shift of a nonlinear oscillator in (linear) response to a perturbation applied at specific phases of the limit cycle. The phase of the unperturbed system is defined with respect to some event, reoccurring at a specific phase. An interesting subclass of nonlinear oscillators to which this theory can be applied are tonically firing integrate-and-fire (IF) neurons subject to weak noise, as considered here. The phase $\tau$ of the oscillator is then defined with respect to the spiking event that, in the deterministic case, occurs with period $T^*$ at times $t_{i-1}$, $t_i$, $t_{i+1}$ and so on. This period can be normalized to 1 or $2\pi$ depending on the context. However, here we interpret $\tau \in [0, T^*]$ as a relative time since the last spike and abstain from normalizing the phase.

In order to calculate the PRC, consider an IF model that is subject to a *weak* perturbation that instantaneously shifts the voltage variable $v$ by some small $\epsilon$ at time $\tau$. This can be realized by applying a delta kick to the dynamics of the neuron model $\dot{v} = f_0(v, \mathbf{w}) + \mu - a + \epsilon\delta(t - (t_{i-1} + \tau))$ (with a reference spike at time $t_{i-1}$). The PRC $Z(\tau)$ measures the shift of the subsequent spike time $t_i(\tau, \epsilon) = t_{i-1} + T_i(\tau, \epsilon)$ or equivalently deviation of the corresponding ISI $\delta T_i(\tau, \epsilon) = T_i(\tau, \epsilon) - T^*$ due to that delta kick applied at "phase" $\tau$

$$Z(\tau) = -\lim_{\epsilon \to 0} \frac{\delta T_i(\tau, \epsilon)}{\epsilon}, \tag{16}$$

The sign on the right hand side is chosen so that a positive PRC is obtained if a positive kicks to the voltage variable $\epsilon > 0$ leads to a shortened ISI $\delta T < 0$.

**Adjoint method.** A useful approach to calculate the PRC analytically is provided by the adjoint method. The PRC satisfies the adjoint equation [40, 42]

$$\dot{\mathbf{Z}}(t) = -A^T \mathbf{Z}(t), \tag{17}$$

where $\mathbf{Z}(t)$ is a set of $N + 2$ functions

$$\mathbf{Z}(t) = [Z(t), Z_{w_1}(t), \ldots Z_{w_N}(t), Z_a(t)]^T. \tag{18}$$

Each component quantifies the linear response of the spike time to a perturbation of the limit cycle in the corresponding variable of the model. In particular, this implies that we can also

calculate the phase-response with respect to perturbations of the auxiliary or adaptation variables. The matrix $A(t)$ is the Jacobian of the considered IF model evaluated at the $T^*$-periodic limit cycle solution $\mathbf{X}_0(t) = [v_0(t), \mathbf{w}_0(t), a_0(t)]^T$

$$A(t) = \begin{pmatrix} \partial_v f_0 & \partial_{w_1} f_0 & \dots & \partial_{w_N} f_0 & -1 \\ \tau_1^{-1} \partial_v f_1 & \tau_1^{-1} \partial_{w_1} f_1 & \dots & \tau_1^{-1} \partial_{w_N} f_1 & 0 \\ \vdots & \vdots & \ddots & \vdots & \vdots \\ \tau_N^{-1} \partial_v f_N & \tau_N^{-1} \partial_{w_1} f_N & \dots & \tau_N^{-1} \partial_{w_N} f_N & 0 \\ 0 & \dots & \dots & 0 & -\tau_a^{-1} \end{pmatrix} \tag{19}$$

with the normalization

$$\mathbf{Z}(t)\dot{\mathbf{X}}_0(t) = Z(t)\dot{v}_0(t) + \mathbf{Z}_w(t)\dot{\mathbf{w}}_0(t) + Z_a(t)\dot{a}_0(t) = 1 \tag{20}$$

and the end (instead of initial) conditions (see [39])

$$Z_{w_1}(T^*) = \dots Z_{w_N}(T^*) = Z_a(T^*) = 0, \tag{21}$$

which, together with Eq (20), implies the condition

$$Z(T^*) = \dot{v}_0(T^*)^{-1}. \tag{22}$$

Solving the above differential equation for $\mathbf{Z}(t)$ with these conditions either analytically (in simple cases) or, for the general case, numerically, provides the PRC with respect to perturbations in the voltage, which is the first component of the vector $\mathbf{Z}(t)$.

## Assumptions for the analytical approximations

We require that the deterministic neuron model exhibits a unique limit-cycle solution with a finite period $T^*$, i.e. the neuron is tonically firing in the absence of noise. Furthermore, following [39] we assume that the effect of the noise sources is weak such that the deviation of the ISI from the deterministic interval $\delta T_i = T_i - T^*$ is small and can be described by the PRC (see above). First of all, this weak-noise condition allows to approximate the average by the deterministic ISI $\langle T_i \rangle \approx T^*$. More rigorous approaches to the mean ISI (or, equivalently, to the mean first-passage time of the voltage going from reset to threshold values) have been pursued for white [107, 108] and colored [29, 48, 109, 110] noise, but are not considered here.

Although in the weak noise limit the mean deviation of the ISI vanishes, correlations between individual ISI deviations do not and the SCC can be approximated by

$$\rho_k \approx \langle \delta T_i \delta T_{i+k} \rangle / \langle \delta T_i^2 \rangle. \tag{23}$$

This is indeed only an approximation due to the assumption that the deviation from the mean is equal to the deviation from the deterministic interval, $T_i - \langle T_i \rangle \approx T_i - T^* = \delta T_i$, which is of course implied by the afore mentioned assumption that the noise does not change the mean ISI.

In the succeeding sections we calculate the SCC for a general nonlinear stochastic IF model with spike-triggered adaptation and driven by a combination of white and colored noise. It simplifies the analytical treatment to first address the special case of a neuron *without adaptation* but with white and colored noise, which is what we do in the next subsection.

## Correlation coefficient for IF models without adaptation

We first demonstrate how the SCC for a non-adaptive stochastic IF model driven by correlated and white noise can be calculated, i.e. in a first step we explicitly exclude the adaptation which can be easily achieved by setting $\Delta = 0$.

Deviations from the mean ISI can be calculated via the PRC $Z(\tau)$, which quantifies how a small displacement of the voltage variable $v$ due to a perturbation $u_i(\tau) = \epsilon\delta(\tau - \tau_0)$, applied a specific "phase" $\tau_0 \in [0, T^*]$ after the last spike time $t_{i-1}$, advances $(Z(\tau_0)u_i(\tau_0) > 0)$ or delays $(Z(\tau_0)u_i(\tau_0) < 0)$ the next spike time $t_i$. In general, a perturbation will affect the ISI deviation over the entire time window (see Fig 1) according to

$$\delta T_i = -\int_0^{T^*} d\tau Z(\tau)u_i(\tau). \tag{24}$$

The PRC $Z(\tau)$ of the deterministic system ($D = \sigma = 0$) is obtained by the adjoint method or numerically, as described above, while the perturbation follows immediately from Eq (2)

$$u_i(\tau) = \eta(t_{i-1} + \tau) + \sqrt{2D}\xi_v(t_{i-1} + \tau). \tag{25}$$

Hence, the deterministic limit cycle is perturbed by two independent processes: a weak OU process $\eta$ and Gaussian white noise $\xi_v$. The deviation of the $i$-th interval is then given by

$$\delta T_i = -\int_0^{T} d\tau \ Z(\tau)[\eta(t_{i-1} + \tau) + \sqrt{2D}\xi_v(t_{i-1} + \tau)], \tag{26}$$

from which we define two random numbers given by the integrals of the weighted noises acting over the $i$-th interval

$$H_{i-1} = \int_0^{T^*} d\tau \ Z(\tau)\eta(t_{i-1} + \tau), \tag{27}$$

$$\Xi_{i-1} = \int_0^{T^*} d\tau \ Z(\tau)\sqrt{2D}\xi_v(t_{i-1} + \tau). \tag{28}$$

Taking products of deviations of intervals that are lagged by an integer $k$, permits to calculate the covariance of intervals needed in the Eq (24) for the SCC:

$$\langle \delta T_i \delta T_{i+k} \rangle = \langle H_i H_{i+k} + \Xi_i \Xi_{i+k} \rangle \tag{29}$$

$$\approx \int_0^{T^*} \int_0^{T^*} d\tau d\tau' \ Z(\tau)Z(\tau')[C_\eta(kT^* + \tau' - \tau) + 2DC_\xi(kT^* + \tau' - \tau)] \tag{30}$$

$$= \int_0^{T^*} \int_0^{T^*} d\tau d\tau' \ Z(\tau)Z(\tau')[\sigma^2 e^{-\frac{|kT^* + \tau' - \tau|}{\tau_\eta}} + 2D\delta(kT^* + \tau' - \tau)]. \tag{31}$$

where we approximated $t_{i+k} - t_i \approx kT^*$ and used the correlation functions of the noise sources, e.g. $C_\eta(\Delta t) = \langle \eta(t_i)\eta(t_i + \Delta t) \rangle$. Note that mixed terms do not contribute since $\eta$ and $\xi_v$ are independent processes. Because of the presence of $\delta$-correlated white noise it is convenient to distinguish between the calculation of the covariance between distinct intervals ($k \geq 1$, here the white noise correlation function does *not* contribute) and the calculation of the variance

$(k = 0)$:

$$\langle \delta T_i \delta T_{i+k}\rangle \approx \begin{cases} \int_0^{T^*}\int_0^{T^*} d\tau d\tau'\, Z(\tau)Z(\tau')\left[\sigma^2 e^{-\frac{|\tau'-\tau|}{\tau_\eta}} + 2D\delta(\tau'-\tau)\right], & k = 0 \\[2ex] \int_0^{T^*}\int_0^{T^*} d\tau d\tau'\, Z(\tau)Z(\tau')\sigma^2 e^{-\frac{|T^*+\tau'-\tau|}{\tau_\eta}}\beta^{k-1}, & k > 0, \end{cases} \tag{32}$$

where $\beta = \exp(-T^*/\tau_\eta)$. Intuitively this distinction implies that the white noise source does increase the variance of the ISIs but does not introduce correlations among the intervals (at least, in the absence of adaptation). We note that the variance of the interspike interval in the case of purely white noise (Eq 32 with $k = 0$ and $\sigma^2 = 0$) reduces to $\langle \delta T_i^2\rangle = 2D\int_0^{T^*} d\tau Z(\tau)^2$, which agrees with an expression derived by Ermentrout et al. [111] (see the unnumbered equation below their Eq 10).

From Eqs (32) and (23) the SCC can be calculated in terms of the correlation functions, yielding for a general PRC an expression in terms of double integrals that have to be evaluated numerically. However, following [47] we choose to express the SCC in the Fourier domain, using the Wiener-Khinchin theorem $C(\Delta t) = \int_{-\infty}^{\infty} d\omega\, S(\omega)e^{i\omega\Delta t}$ to substitute the correlation functions, and the finite Fourier transform of the PRC $\tilde{Z}(\omega) = T^{*-1}\int_0^{T^*} d\tau\, Z(\tau)e^{i\omega\tau}$. Eq (32) then becomes

$$\langle \delta T_i \delta T_{i+k}\rangle \approx \begin{cases} \int d\omega |\tilde{Z}(\omega)|^2[2\tau_\eta\sigma^2(1+\omega^2\tau_\eta^2)^{-1} + 2D], & k = 0, \\[2ex] \int d\omega |\tilde{Z}(\omega)|^2[2\tau_\eta\sigma^2(1+\omega^2\tau_\eta^2)^{-1}e^{-i\omega T^*}]\beta^{k-1}, & k > 0. \end{cases} \tag{33}$$

The SCC reads (as stated in the main part)

$$\rho_{k,\eta} = \frac{\int d\omega |\tilde{Z}(\omega)|^2(1+\omega^2\tau_\eta^2)^{-1}e^{-i\omega T^*}}{\int d\omega |\tilde{Z}(\omega)|^2[(1+\omega^2\tau_\eta^2)^{-1} + D/(\tau_\eta\sigma^2)]}\beta^{k-1}. \tag{34}$$

For $D = 0$ this expression agrees with the one derived by Schwalger et al. [47]. Remarkably, Eq (34) is the only place where the intensity of the white noise and the variance of the colored noise appear in our theory. Note that the limit $D \to 0$, $\sigma^2 \to 0$ is well defined for a fixed ratio of noise intensities, $D/(\tau_\eta\sigma^2)$.

## Derivation of the general correlation coefficient

To calculate the SCC for the full system we pursue a similar strategy as done by Ref. [39]. We find a relation between ISI deviations and perturbations and another relation between ISI deviations and adjacent peak adaptation values. These two relations allow i) to express $\rho_k$ by the covariance of the peak adaptation values $c_k$ and ii) to find a stochastic map for the peak adaptation values from which this covariance can be calculated.

First ISI deviations can be calculated again via the PRC $Z(\tau)$ as described above. To this end we separate limit cycle dynamics from noise induced perturbations. In this case the perturbation

$$u_i(\tau) = -\delta a_{i-1}e^{-\tau/\tau_a} + \eta(t_{i-1}+\tau) + \sqrt{2D}\xi_\nu(t_{i-1}+\tau), \tag{35}$$

acting over one ISI $T_i$ is found by rewriting Eq (2) as

$$\dot{\nu} = f_0(\nu, \mathbf{w}) + \mu - a^* e^{-\tau/\tau_a} + u_i(\tau), \tag{36}$$

with the deterministic peak adaptation value $a^* = (\Delta/\tau_a)/(1 - e^{-T^*/\tau_a})$ and the deviation from

it $\delta a_{i-1} = a(t_{i-1}) - a^*$ [$a(t_i)$ taken right *after* the incrementation]. Here, the deterministic limit cycle is not only perturbed by a weak OU process $\eta$ and Gaussian white noise $\xi_v$ but also by a small deviation in the peak adaptation value $\delta a_{i-1}$. Combining Eqs (24) with (35) and shifting the index for notational convenience, we obtain

$$\delta T_{i+1} = \int_0^{T^*} d\tau \; Z(\tau)(\delta a_i e^{-\tau/\tau_a} - \eta(t_i + \tau) - \sqrt{2D}\xi(t_i + \tau)). \tag{37}$$

In contrast to the case considered in the previous section interval correlations can not be calculated from this equation directly because correlations among peak adaptation values $\langle \delta a_i \delta a_{i+k} \rangle$ are unknown. Moreover the peak adaptation value and, for example, colored noise are not independent of each other $\langle \delta a_i H_{i+1} \rangle \neq 0$.

Instead we derive a second expression for the interval deviations by relating adjacent peak adaptation values $a_i = a(t_i)$, $a_{i+1} = a(t_{i+1})$, that will generally deviate from the deterministic value, $\delta a_i = a_i - a^* \neq 0$. Since the adaptation is spike triggered, its time course over one ISI is deterministic. The only non-deterministic quantity relating the two adjacent peak adaptation values is the length of the corresponding ISI $T_{i+1}$

$$a_{i+1} = a_i e^{-T_{i+1}/\tau_a} + \Delta/\tau_a. \tag{38}$$

Deviations $\delta T_i$ and $\delta a_i, \delta a_{i+1}$ from their deterministic values can be related by linearizing Eq (38)

$$\delta T_{i+1} = \frac{\tau_a}{a^*}(\delta a_i - e^{T^*/\tau_a}\delta a_{i+1}), \tag{39}$$

i.e. two adjacent peak adaptation values give us knowledge about the deviation of the interval in between. Inserting Eq (39) into (23 we find an expression for the SCC in terms of the covariances $c_k = \langle \delta a_i \delta a_{i+k} \rangle$ of the peak adaptation values:

$$\rho_k \approx \frac{(\alpha^2 + 1)c_k - \alpha(c_{k-1} + c_{k+1})}{(\alpha^2 + 1)c_0 - 2\alpha c_1} \tag{40}$$

with $\alpha = \exp(-T^*/\tau_a)$ that contains the adaptation time scale. Next we determine the covariance from a stochastic map that is derived by combining Eqs (37) and (39)

$$\delta a_{i+1} = (\alpha v)\delta a_i + \frac{\alpha a^*}{\tau_a}(H_i + \Xi_i), \tag{41}$$

where the linear response to adaptation is described by

$$v = 1 - \frac{a^*}{\tau_a}\int_0^{T^*} d\tau \; Z(\tau)e^{-t/\tau_a}, \tag{42}$$

while the linear responses to colored and white noise $H_i, \Xi_i$ are defined as in the proceeding section [Eqs (27) and (28)] Note that $v$ is constant, which reflects the fact that the adaptation dynamics does neither include noise nor subthreshold adaptation. In contrast, $H_i$ and $\Xi_i$ are random numbers. Furthermore the stability of the stochastic map Eq (41) requires $|\alpha v| < 1$.

Recursively applying the map to itself yields an relation not only between adjacent peak adaptation values, i.e. lag 1, but any lag k

$$\delta a_{i+k} = (\alpha v)^k \delta a_i + \frac{\alpha a^*}{\tau_a}\sum_{n=1}^{k}(\alpha v)^{k-n}(\Xi_{i+n-1} + H_{i+n-1}). \tag{43}$$

The covariance is obtained by multiplying Eq (43) with $\delta a_i$ and applying the average

$$c_k = (\alpha v)^k \langle \delta a_i^2 \rangle + \frac{\alpha a^*}{\tau_a} \sum_{n=1}^{k} (\alpha v)^{k-n} \langle \delta a_i H_{i+n-1} \rangle. \tag{44}$$

Note that $\langle \delta a_i \Xi_{i+n-1} \rangle = 0$ for $n > 0$ because $\Xi_i$ only contains the uncorrelated white noise *after* the spike time $t_i$ to which the peak adaptation value $\delta a_i$ belongs. The remaining terms of the sum can be simplified using the exponential decay of the averaged OU process

$$\langle \delta a_i H_{i+k} \rangle = \beta^k \langle \delta a_i H_i \rangle, \tag{45}$$

with $\beta = \exp(-T^*/\tau_\eta)$ and $k \geq 0$. The resulting geometric sequence can be summed, yielding the covariance

$$c_k = (\alpha v)^k \langle \delta a_i^2 \rangle + \frac{(\alpha v)^k - \beta^k}{\alpha v - \beta} \frac{\alpha a^*}{\tau_a} \langle \delta a_i H_i \rangle. \tag{46}$$

The so obtained covariance for any lag $k$ is determined by the variance $\langle \delta a_i^2 \rangle$, covariance $\langle \delta a_i H_i \rangle$ (at lag $k = 0$), and two prefactor that exclusively carry the dependence on $k$. To determine the SCC from $c_k$ according to Eq (40), we use the stochastic map to calculate the ratio $\langle \delta a_i^2 \rangle / \langle \delta a_i H_i \rangle$ (it turns out that this ratio is all we have to know). We obtain a first equation by squaring the map and averaging

$$\langle \delta a_{i+1}^2 \rangle = (\alpha v)^2 \langle \delta a_i^2 \rangle + \frac{2 \alpha^2 v a^*}{\tau_a} \langle \delta a_i H_i \rangle + \left( \frac{\alpha a^*}{\tau_a} \right)^2 (\langle H_i^2 \rangle + \langle \Xi_i^2 \rangle); \tag{47}$$

for further use below we note that in the stationary case $\langle \delta a_{i+1}^2 \rangle = \langle \delta a_i^2 \rangle$. A second equation is obtained by multiplying the map with $H_{i+1}$ and averaging

$$\langle \delta a_{i+1} H_{i+1} \rangle = \alpha v \langle \delta a_i H_{i+1} \rangle + \frac{\alpha a^*}{\tau_a} \langle H_i H_{i+1} \rangle = \frac{\alpha a^*}{\tau_a} \frac{\langle H_i H_{i+1} \rangle}{1 - \alpha v \beta}. \tag{48}$$

Here we used Eq (45) and the fact that terms containing $\Xi_i$ drop out for the reason discussed above. Combining Eqs (47) and (48) yields the desired relation

$$\frac{\tau_a}{\alpha a^*} \frac{\langle \delta a_i^2 \rangle}{\langle \delta a_i H_i \rangle} = \frac{2 \alpha v}{1 - (\alpha v)^2} + \frac{1 - \alpha v \beta}{1 - (\alpha v)^2} \frac{\langle H_i^2 \rangle + \langle \Xi_i^2 \rangle}{\langle H_i H_{i+1} \rangle} \tag{49}$$

The remaining terms $\langle \Xi_i^2 \rangle$, $\langle H_i^2 \rangle$ and $\langle H_i H_{i+1} \rangle$ are the known correlation functions of white and colored noise and can be calculated as done in the preceeding section.

We can now calculate the SCC by combining Eq (40) with 46, divide both numerator and denominator by $\langle \delta a_i H_i \rangle$, apply Eq (49) and find the result presented in the main part

$$\rho_k = \left( \frac{A}{C} \right) \rho_{k,a} + \left( \frac{B}{C} \right) \rho_{k,\eta}. \tag{50}$$

with coefficients

$$A = 1 + \frac{(1 + (\alpha v)^2 - 2\alpha v\beta)}{\alpha v - \beta}\rho_{1,\eta} - \alpha v\beta, \tag{51a}$$

$$B = \frac{(1 - (\alpha v)^2)(1 - \alpha\beta)(\alpha - \beta)}{(1 + \alpha^2 - 2\alpha^2 v)(\alpha v - \beta)}, \tag{51b}$$

$$C = 1 + 2\rho_{1,a}\rho_{1,\eta} - \alpha v\beta \tag{51c}$$

$$\alpha = e^{-\frac{T^*}{\tau_a}}, \quad \beta = e^{-\frac{T^*}{\tau_\eta}}, \quad v = 1 - \frac{a^*}{\tau_a}\int_0^{T^*} d\tau \; Z(\tau)e^{-\tau/\tau_a}. \tag{51d}$$

and specific correlation coefficients

$$\rho_{k,a} = -\frac{\alpha(1 - \alpha^2 v)}{1 + \alpha^2 - 2\alpha^2 v}(1 - v)(\alpha v)^{k-1}, \tag{52a}$$

$$\rho_{k,\eta} = \frac{\int d\omega |\tilde{Z}(\omega)|^2 S_{ou}(\omega)e^{-i\omega T^*}}{\int d\omega |\tilde{Z}(\omega)|^2 [S_{ou}(\omega) + 2D]}\beta^{k-1}. \tag{52b}$$

As easily verified, the main result Eq (50) is consistent in different limit cases. Switching off the colored noise ($\sigma = 0$ implying $S_{ou}(\omega) \equiv 0$) yields $\rho_k = \rho_{k,a}$. Similarly, setting $\Delta = 0$ (vanishing adaptation) results in $v = 1$ and $\rho_k = \rho_{k,\eta}$.

## Coefficient of variation

Finally, we derive the coefficient of variation for the full system which follows a similar scheme as the derivation of the correlation coefficient presented above. Again, we use the assumption that the mean ISI can be approximated by the deterministic ISI. This allows us to express the CV in terms of the derivation $\delta T_i$ of the spike time and the deterministic period $T^*$

$$C_V^2 = \frac{\langle(T_i - \langle T_i\rangle)^2\rangle}{\langle T_i\rangle^2} \approx \frac{\langle\delta T_i^2\rangle}{T^{*2}}. \tag{53}$$

Eq (39) can be used to relate the variance of the ISI to the variance and co-variance of the adaptation variable

$$\langle\delta T_i^2\rangle = \left(\frac{\tau_a}{a^*}\right)^2(\langle\delta a_i^2\rangle - 2\alpha^{-1}\langle\delta a_{i+1}\delta a_i\rangle + \alpha^{-2}\langle\delta a_i^2\rangle). \tag{54}$$

The latter variance and co-variance can be traced back to the statistics of the uncorrelated and correlated Gaussian random numbers, $\Xi_i$ and $H_i$, respectively. Specifically, we derive three relations from the stochastic map Eq (41) that are related to Eqs (47) and (46) at lag 1 and

Eq (48)

$$\langle \delta a_i^2 \rangle = \frac{(\alpha a^*)^2}{\tau_a^2(1-(\alpha v)^2)}[\langle H_i^2 \rangle + \langle \Xi_i^2 \rangle] + \frac{2\alpha^2 v a^*}{\tau_a(1-(\alpha v)^2)}\langle \delta a_i H_i \rangle. \tag{55}$$

$$\langle \delta a_{i+1} \delta a_i \rangle = (\alpha v)\langle \delta a_i^2 \rangle + \frac{\alpha a^*}{\tau_a}\langle \delta a_i H_i \rangle. \tag{56}$$

$$\langle \delta a_{i+1} H_{i+1} \rangle = \frac{\alpha a^*}{\tau_a(1-\alpha v\beta)}\langle H_i H_{i+1} \rangle. \tag{57}$$

Substituting these expressions into Eq (53) and dividing by $T^{*2}$ yields the following expression:

$$C_V^2 = \frac{1+\alpha^2-2\alpha^2 v}{T^{*2}(1-(\alpha v)^2)}[\langle H_i^2 \rangle + \langle \Xi_i^2 \rangle] - \frac{2\alpha(1-\alpha^2 v)(1-v)}{T^{*2}(1-\alpha v\beta)(1-(\alpha v)^2)}\langle H_i H_{i+1} \rangle. \tag{58}$$

We recall that $H_i$ and $\Xi_i$ are the integrated and PRC weighted colored and white noise sources and that correspondingly the terms above are found to be

$$\langle H_i^2 \rangle + \langle \Xi_i^2 \rangle = \int_0^{T^*}\int_0^{T^*} d\tau d\tau' \, Z(\tau)Z(\tau')\left[\sigma^2 e^{-\frac{|\tau'-\tau|}{\tau_\eta}} + 2D\delta(\tau'-\tau)\right] \tag{59}$$

$$\langle H_i H_{i+1} \rangle = \int_0^{T^*}\int_0^{T^*} d\tau d\tau' \, Z(\tau)Z(\tau')\sigma^2 e^{-\frac{|T^*+\tau'-\tau|}{\tau_\eta}} \tag{60}$$

in strict analogy to the considerations in section Correlation coefficient for IF models without adaptation.

We stress again that our theory is based on the linear response function of a phase model, namely the phase-response curve, a description that holds true only in leading order of the perturbation amplitude (e.g. $\sigma$ in the case of purely colored noise). Therefore, in this linear framework, deviations in the phase and in consequence in the interspike-interval caused by perturbations with amplitude $\sigma$ can meaningfully be described only up to linear order in $\sigma$. This also implies that averages of products of intervals depend in leading order on $\sigma^2$. Any extension into the realm of higher-order terms must be based on nonlinear response functions.

## Details on the numerical integration of stochastic equations

For the numerical integration of Eq (2), a stochastic Euler scheme was used with step size $\Delta t = 10^{-5}$. The simulation was terminated after $5 \cdot 10^4$ to $10^5$ spikes; for all curves shown the standard error of the estimated mean of $\rho_1$ was below 0.009.

To reduce transient effects, the initial conditions were chosen to be on the deterministic limit cycle, specifically $v_0 = v_R$, $\mathbf{w}_0 = \mathbf{w}_R$, $a_0 = a^*$, $\eta_0 = 0$. The deterministic peak adaptation value $a^*$ can be calculated from the deterministic period via $a^* = [1 - \exp(-T^*/\tau_a)]\Delta/\tau_a$ or, if $T^*$ is analytically inaccessible, directly from integrating the noiseless system. Of course, the specific initial values of the noise variable and adaptation variable in the stochastic system impose a transient (non-stationary) behavior in any measured statistics. However, due to the large number of subsequent ISIs, the transients have little effect on the considered statistics. Indeed, for selected parameter sets, we have verified that the initial values do not matter for the measured SCC.

## Acknowledgments

We thank Tilo Schwalger for useful discussions on the topic of this manuscript.

## Author Contributions

**Conceptualization:** Lukas Ramlow, Benjamin Lindner.

**Data curation:** Lukas Ramlow.

**Formal analysis:** Lukas Ramlow, Benjamin Lindner.

**Funding acquisition:** Lukas Ramlow, Benjamin Lindner.

**Investigation:** Lukas Ramlow, Benjamin Lindner.

**Methodology:** Lukas Ramlow, Benjamin Lindner.

**Project administration:** Lukas Ramlow, Benjamin Lindner.

**Resources:** Lukas Ramlow, Benjamin Lindner.

**Software:** Lukas Ramlow.

**Supervision:** Lukas Ramlow, Benjamin Lindner.

**Validation:** Lukas Ramlow, Benjamin Lindner.

**Visualization:** Lukas Ramlow, Benjamin Lindner.

**Writing – original draft:** Lukas Ramlow, Benjamin Lindner.

**Writing – review & editing:** Lukas Ramlow, Benjamin Lindner.

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
