## [Decision Letter · Decision Letter 0]

26 Aug 2020

Dear Mr. Ramlow,

Thank you very much for submitting your manuscript "Interspike interval correlations in neuron models with adaptation and correlated noise" for consideration at PLOS Computational Biology.

As with all papers reviewed by the journal, your manuscript was reviewed by members of the editorial board and by several independent reviewers. In light of the reviews (below this email), we would like to invite the resubmission of a significantly-revised version that takes into account the reviewers' comments. I'll just note here that I'm not terribly concerned that the manuscript is too focused on a "technicality." This journal welcomes technical work provided it is relevant to biology. That being said, to the extent you can make the paper inviting for a broader audience, so much the better.

We cannot make any decision about publication until we have seen the revised manuscript and your response to the reviewers' comments. Your revised manuscript is also likely to be sent to reviewers for further evaluation.

Sincerely,

Samuel J. Gershman

Deputy Editor

PLOS Computational Biology

Reviewer's Responses to Questions

**Comments to the Authors:**

Reviewer #1: The present manuscript is an analytical modelling study supported by numerical simulations. It describes the second-order statistics for the output spike trains of multi-dimensional integrate-and-fire neuron models in the presence of both a single exponential adaptation current and a correlated noise current in the form of an Ornstein-Uhlenbeck process.

Overall, I think the manuscript reads well and presents the content in an accessible fashion. The authors provide a lot of very insightful intuition about their findings, which, in my eyes, makes the manuscript particularly suitable for publication in PLOS Computational Biology. The authors also start first with special cases and then go on to derive and explain their main theory, which adds to the accessibility of the manuscript.

Most importantly though, a recent experimental finding is explained by new theoretical results in the manuscript, which should make the paper an excellent match for PLOS Computational Biology.

I think the claim that no present model (i.e. integrate-and-fire models without both adaptation and correlated noise input) can generate a correlation structure as presented in Fig. 2(d) is substantiated. Also, the result that a single-exponential adaptation current in a multidimensional integrate-and-fire model (shown in Figs. 5(c) and 6) can induce strong positive serial correlations is very interesting and the interplay of correlated noise and adaptation has never been studied in this setup. Conversely, the authors also show that in the absence of adaptation, a positively correlated noise process can induce negative correlations in a similar model (Fig. 6(b)). As these effects are quite subtle, I would expect that they could be used in a model selection setup to find the best (maybe even unique) minimal model for a given set of noisy spike train observations. As such, the results could prove to be valuable for a wider neuroscience audience.

I mostly have minor comments and suggestions, but before that, I would like to start by suggesting a few more general points to consider.

1. Wouldn't it add to the readability of the manuscript if in Eq. 2, N was set to 1? In the manuscript, the authors only ever look at N=0 or N=1, so maybe the notation could be eased accordingly. If not, I suggest to clearly indicate that j=1,..,N in Eq. 2b.

2. Related to 1., why is the most general case covered by Eq. 2 not treated with an example in the manuscript? In figs. 2-4 and 7, N=0 (in Fig. 2, also D=0), whereas in figs. 5-6, D=0, but N=1. Can the authors please comment on this choice?

3. I suggest to reference the paper [47] in list of references contained in the manuscript earlier, and then mention in the Introduction (for example starting on p.3, l.57) that the main new theoretical finding of the manuscript is backed up by experimental findings. The paper [47] seems to be mainly a theoretical study, could the authors additionally cite the original experimental papers on which the study [47] builds?

4. It would be good if the authors stated that stationarity of spike trains is assumed to compute the SCC using Eq. 1. Related to this, could the authors please give additional details on their simulation setup (timestep and algorithms for the numerical integration, sample size to estimate the SCC, sample standard deviations of the SCC across realizations and burn-in period durations). This could be done briefly in the Methods section of the paper.

/////////////////////////////

Minor comments

/////////////////////////////

1. Concerning the Author Summary (p.1, l.1 ff.), it might be good to mention why SCCs are an important and useful quantity (for example for information transmission and weak signal detection; these considerations are briefly mentioned in the Introduction section and presented in more detail in the Discussion and Conclusion section (p.10)) also with a view on experiments, because, in my opinion, the Author Summary currently is a bit too technical for a general audience of computational neuroscientists.

2. Related to fig. 1: Where is u in Eqs. 2a-d? I understand that Fig. 1 serves to introduce the concept of a phase resetting curve (it becomes clearer after reading the Methods section), but I find the introduction of the new variable u confusing.

3. The abbreviation "IF" is introduced multiple times in the manuscript.

4. On p.3, l.41, the authors say that spike-frequency adaptation is '...exhibited by almost any type of neuron...'. I don't think this is true in such generality because spike-frequency adaptation seems to be a property mainly of principal (pyramidal) cells.

5. p.3, l.47: What is 'adaptation channel noise'? Is it a noisy current caused by the stochastic opening and closing of ion channels mediating an adaptation current? Please explain.

6. p.3, l.57: When mentioning 'novel patterns of interval correlations', it is not clear whether they are theoretical predictions of experimental results (this becomes clearer later in the manuscript). I suggest to point this out already here to make it clear. Related to this, it might also be worth mentioning it in the presentation of the results for the QIF model on p.4, l.98 ff.

7. In Eq. 2c, is there a reason why the adaptation jump size is chosen as Delta/tau_a and not Delta, as in previous studies (e.g. reference [38], Schwalger and Lindner 2013, Front Comp Neurosci)?

8. After their main result (Eq. 3), could the authors comment on how the dynamics of the auxiliary variables w (given by f_j in Eq. 2b) influences the result for the SCC? I assume mainly/exclusively through the phase resetting curve Z. Is this true?

9. p.4, l.109 ff.: It would be good to point out that 'weak noise' means small D in Eq. 2a, maybe even already after the presentation of the model equations on p. 3. That is, weak noise mean 'weak white noise' in this paper.

10. Is the limit D=0 for rho_k,a in Eq. 3 well-defined? I think it is, because alpha, beta and nu (Eq. 4d) are independent of D and so is rho_k,a (Eq. 5). However, A and C are dependent on D via rho_1,eta. Does this pose a problem? Or, differently put, what is the SCC in an IF model with adaptation, but without white and colored noise (D=sigma=0)? It would be great if the authors commented on it, as this case seems to be treated in Fig. 7(a).

11. A (Eq. 4a) and C (Eq. 4c) depend on rho_1,eta and hence also on properties of the colored noise, whereas the authors say that '...the dependence on the lag k is carried exclusively by the two specific SCCs...'. From the formulas, this is of course true, but maybe it is misleading because it might be misunderstood to mean that the first term in Eq. 3 does not depend on any property of eta and similarly for the second term.

12. p.5, l.121: '...a structure that cannot be generated by a single geometric series...'- Maybe it is good to point out that SCC dependencies on k that are generated by single geometric series need to be either monotonically decaying/ increasing (base >0, depending on sign of prefactor) or oscillatory and decaying (base <0) functions of k for completeness and clarity.

13. p.6, l.135: Could the authors explain the origin of the 'exponential function' that envelops correlation coefficients? Does this envelope come from rho_k, eta?

14. p.6, l. 142: The expression given for Z(T*) does not look correct. The bracket should be closed after v_T; what is 'a*' and why is there a Delta/tau_a appearing? I understand that T* is the deterministic period of the oscillator, which can be obtained by integrating Eq. 2 for eta=D=0 from v_R to v_T and then solving for T*. Please explain.

15. p.7, l. 149: What is E? It must be related to Eq. 5. Also, on l. 151, an equation reference went awry.

16. Related to the explanations on p.7, l. 149 ff.: Can rho_k,eta (Eq. 6) ever be negative? I would assume not, but on l.152, the authors write: '...and possesses a non-negative prefactor for positive PRCs...'. The PRC enters in Eq. 6 only as the squared magnitude of its Fourier transform, which should always be positive, no matter the sign of Z? Please explain.

17. p.7, l. 156: I think the patterns should be described as 'exponentially increasing or oscillatory', as the SCC at lag 1 is the most negative one (the SCC goes to zero for large lags in the presence of only an adaptation current, like the blue curve in fig. 3(a)). Similarly, this could be changed in the caption of Fig. 3.

18. p.7, l.161: Can the authors briefly explain why strong and weak adaptation are parametrized by the sign of nu?

19. p.8, l.188: The SCC is not positive everywhere in Fig. 5(c), see for example the light green curve. Please explain and correct. I also find it confusing (clearly though, this a novel non-trivial finding) that there is no clear dependence on tau_eta in Fig. 5c; for example, the yellow and the green curve differ by only one order of magnitude in tau_eta but show completely different behaviors, whereas the two curves with the lowest values for tau_eta are encased by the yellow and green curve. I think this is described in the caption of Fig. 5.

20. p.8, l.194-95: Maybe it would be good to point out that the perturbation u (in terms of Fig. 1) is now negative, so it will advance firing if it acts on the beginning of the next interval where Z is negative.

21. p.8, l. 220 and caption of Fig. 7: The authors should describe again which model is considered in detail, for example by noting down the equations or reference earlier models in the manuscript.

22. section beginning on p.9, l.219: I suggest to give some more details on the relation of the abstract IF model with a model for a slow stochastic population of adaptation channels, maybe a small sketch is helpful in this regard.

23. related to p.9., l.233 (and also earlier in the manuscript, e.g. p.6, l.134): I find it confusing to call the non-monotonic dependence of the SCC on k 'oscillatory'. Oscillations usually have a time component to them. Maybe it is better to call the behavior 'sign-changing' or simply 'non-monotonic'?

24. p.9, l.243: 'tau' should be 'tau_c'.

25. p.9, l.244: What is the 'stochasticity of the adaptation' in the case of weak white noise? Please explain.

26. p. 11, equation before l. 336: There should be a reference given for this classical equation.

27. Why do the authors expect that their weak-noise theory also works for stronger noise, as stated in the Discussion and Conclusion section? As far as I can see, it seems that the maximal value considered in the manuscript is D =0.05 in Fig. 7. Please explain.

28. It would be good to give the numerical values for the deterministic period T* in figs. 5-7 for clarity.

29. Discussion beginning on p.12, l.364: I don't understand how the two recently proposed techniques to compute the phase of a stochastic oscillator could be used for the computations of the present manuscript, because in my understanding, these two approaches are purely numerical and the techniques in the manuscript require analytical expressions for the PRC. Please explain.

////////////////////////////////

////////////////////////////////

Comments related to spelling and grammar:

p.2, caption of Fig. 1: 'integrated' instead of 'integrate'.

p.2, l.30: there should be no comma after 'although'.

p.2, l.33: Missing 'e' in 'therefor'.

p.3, l.45: It should be 'by far' and not 'far'.

p.3, l.51: 'correlation-inducing' instead of 'correlation inducing'

p.6., caption of Fig. 3: 'case shown' should be 'case is shown'.

p.6, l.145: 'brings' instead of 'bring'

p.6, caption of Fig. 4: 'SCCs' should be 'SCC'.

p.7, l.170: 'like' should be 'similar'.

p.8, caption of Fig. 6: 'effected' should be 'affected'.

p.8, l.207: comma missing after 'T*'

p.9, caption of Fig. 7: 'to' should be 'so'.

p.10, l. 285: 'in order predict' should be 'in order to predict'.

p.17, l.436: typos: 'proceeding' should be 'preceeding' and around 'device'.

p.18, l.441: 'ms' should be 'manuscript'.

There is one comma and a full stop missing at the end of the captions for figures 5 and 6.

Reviewer #2: Ramlow and Lindner's work calculates serial correlations between adjacent spikes in a multidimensional integrate-and-fire neuronal model with temporal input correlation, and the spike-triggered spike-frequency adaptation. Their method is based on the standard calculation of the Phase-Response Curve (PRC) for the neuron models in a noise-less setting and then averaging it over a weak noise realization. The method has severe limitations; it requires a substantial mean input far beyond the spiking threshold and feeble noise. Also, adaptation and temporal noise correlation should have both rather long time-constants. Using this method, they report that in both cases of a temporally correlated input and the spike-frequency adaptation, one cannot approximate the inter-spike-interval serial correlation using a geometric series. The other reported technical result is that the inter-spike serial correlations can be negative in a neuronal resonator model due to PRC's shape, while neurons receive a colored noise input in some parameter range.

The present method is a rather standard approach in physics and computational/theoretical neuroscience (see, for example, classical Ermentrout (1996)). Also, I believe the manuscript lacks new results, and its limiting assumption for the calculation is major. Additionally, I do not see how these limited technicalities can be relevant to the border readership of PLoS Computational Biology.

The PRC's application in this setting strongly requires that the noisy input does not alter the spiking frequency of the neural models. This assumption is majorly incorrect in a wide range of physiological parameters where the serial correlations can be relevant (for example, see spiking CV in Chacron et al. 2003). One can see the parameters in Fig.3: The mean input $\\mu$ is set to be 5 or 20 while the temporal variance $\\sigma^2$ of colored noise is set to 0.02, and for white noise, they use 0.001. Similarly, one can see these enormous differences in Fig.4-7. It is unclear to what extent one can expect the theory to fit the simulations.

In higher dimensional models, similar to the resonator example described in Eqs 9a and 9b, the PRC reflects the system's phase along the limit cycle. In these models, one expects that the fluctuation along the orthogonal dimension also contributes in the same order to the serial correlations. However, those contributions are not present in the current analysis on page 8/23.

As a reader of PLoS CB, I'd consider this submission a technicality, and it lacks sufficient new results. Therefore, I advise a specialized journal could be more suitable for this work.

**Have all data underlying the figures and results presented in the manuscript been provided?**

Reviewer #1: Yes

Reviewer #2: Yes

PLOS authors have the option to publish the peer review history of their article (what does this mean?). If published, this will include your full peer review and any attached files.

Reviewer #1: No

Reviewer #2: No
---

## [Decision Letter · Decision Letter 1]

30 Oct 2020

Dear Mr. Ramlow,

Thank you very much for submitting your manuscript "Interspike interval correlations in neuron models with adaptation and correlated noise" (PCOMPBIOL-D-20-00911R1) for consideration at PLOS Computational Biology. As with all papers peer reviewed by the journal, your manuscript was reviewed by members of the editorial board and by several independent peer reviewers. Based on the reports, we regret to inform you that we will not be pursuing this manuscript for publication at PLOS Computational Biology.

The review is attached below this email, and we hope you will find it helpful if you decide to revise the manuscript for submission elsewhere. As you will see, the reviewer makes a strong argument that the manuscript has several critical shortcomings. We are sorry that we cannot be more positive on this occasion. We very much appreciate your wish to present your work in one of PLOS's Open Access publications. 

Thank you for your support, and we hope that you will consider PLOS Computational Biology for other submissions in the future.

Sincerely,

Samuel J. Gershman

Deputy Editor

PLOS Computational Biology

Reviewer's Responses to Questions

**Comments to the Authors:.**

Reviewer #2: Ramlow and Lindner have submitted a revised manuscript that develops a theory to address the auto-correlation observed in interspike-intervals (ISIs) of Leaky-Integrate-and-Fire (LIF) with an additional slow variable (colored noise input and spike-triggered adaptation). The revision certainly improved the manuscript. I acknowledge and appreciate the authors' efforts. However, the revision did not resolve my major concerns.

1- Novelty of the approach

In the first review comment, I indicated that the developed theory (if it is correct) is based on an approach that is standard and does not go beyond previous works methodologically. The authors disagreed. Note that I do not mean that there are no open theoretical questions in the methodology, but in contrast, one can develop standard methods to tackle unsolved problems. However, the authors are silent about those critical issues in using phase-response formalism to compute higher-order spiking statistics. Those issues make the current theory mathematically wrong (See point 2 in this report). Here, I include a few papers using the same methodology to show the approach's historical scope to account for spiking statistics using phase-reduction methods.

Ermentrout et al. (J Comput Neurosci 31, 2011) calculate ISIs distribution's mean and variance for a general multi-dimensional oscillator dynamics with additive Gaussian noise. Their results presented on pages 20 and 21 under the Coefficient of variation are specific examples of this paper. This paper is not cited.

Ota et al. (PLoS One, 2012) calculated higher-order statistics of neural oscillators. Importantly, this work shows spike covariance (eq. 11), a related quantity to the authors' primary results in eq. 24, which must be evaluated up to higher orders in the perturbation expansion. These additional terms are missing in the authors' calculation (see point 2 in this report). This paper is also, unfortunately, not cited.

Another very relevant work, which is missing in the references, is the paper by Teramae et al. (PRL102, 2009). In this paper, they have studied a general oscillator driven by colored noise. This is indeed similar to the model in eq. 2 of this submission. The result here indicates that a noisy oscillator phase's dynamics do have an order-one term orthogonal to the limit-cycle as well, see $k(\\phi)$ in equation 12 therein, and therefore cannot be neglected. This term is missing in the authors' calculation of the serial ISI correlations without any reasoning.

They are plenty of more similar examples in the literature. However, I end this list with a paper by van Vreeswijk (PRL 84, 2000). In that paper, van Vreeswijk shows in a general setting, as a result of a perturbation, the timing of n-th spikes has a higher-order contribution from the corresponding Floquet exponent (eq. 6-9). These contributions are essential for calculating the interspike-intervals serial correlations. Moreover, this work includes the specific case of spike-triggered-adaptation, and it has also derived the additional contributions in perturbation series (see eq. 18 and 19 therein). The work here is highly relevant for authors, but it is overlooked in Ramlow and Lindner's revision.

2- Mathematical correctness

In the first review report, I asked why the contribution of the fluctuation along the orthogonal dimensions to the limit cycle is absent in the calculation. The authors responded by citing a Wilson (PRE, 2019) paper to justify why they neglect higher-order terms. The result by Wilson (PRE, 2019) is an asymptotic result for the phase-amplitude reduction (they are more insightful and accessible papers on this topic; for instance, please look at Wedgwood et al. (J Math. Neurosci 3, 2013), Castejón et al. (J Math Neurosci 3, 2013) and Pérez-Cervera et al. (Chaos 30, 2020)). Wilson (PRE, 2019) projects dimensions orthogonal to the limit-cycle into isostables structure to drive the reduction. By definition, the isostables define as the space that perturbations go back asymptotically to the same limit-cycle. This is irrelevant for the closeby ISIs as the authors study it here. Therefore, this only argument of the authors is not convening, and it is unclear why simulations and mathematical derivations can fit at all in some range.

Moreover, as it is shortly mentioned above in works by van Vreeswijk (PRL84, 2000), Teramae et al. (PRL102, 2009), and specially Ota et al. (PLoS One, 2012) they all show that the higher-order terms have to contribute to higher-order statistics of ISIs. This is an exciting and open problem in the realm of this standard approach, and unfortunately, it is overlooked.

One can understand this intuitively. Let's consider a noiseless system with a periodic solution, and it happily moves along its trajectory. A small stochastic perturbation shifts its time to the first spike as it changes the trajectory shape temporarily. Therefore, it also contributes to the second spike (especially if it has a slower variable in its Cartesian description). The trajectory does not automatically reset to its equilibrium starting point after just one cycle. The shape of the phase-response function only goes back to its steady-state with some non-trivial timescale. This is very nicely explained by Teramae et al. (PRL102, 2009) for a general system with a limit-cycle driven by colored noise (similar to the model that Ramlow and Lindner consider in this manuscript). Teramae et al. (PRL102, 2009) clearly state that the orthogonal direction's fluctuation to the limit cycle remains order-one (see the text above the eq. 3 therein). This is important if one needs to calculate the correlation of two back-to-back interspike-intervals. 

Let me rephrase to make sure that this comment arrives at the authors in the most collegial way. In eq. 19 in the submission, we have only the perturbation along the limit-cycle because we define $u_i(t)=\\epsilon\\delta(\\tau-\\tau_0)$ in that sense. However, in eq. 2 (Cartesian description), the noise also has components along the orthogonal dimension to the limit-cycle as it is written in eq. 2 in Teramae et al. (PRL102, 2009). Therefore, when the limit-cycle is stable, one must introduce an amplitude-relaxation time for it, which remains order-one after the perturbation. Missing the correct transformation makes the perturbation series truncated wrongly. Thus, by looking at the calculation in van Vreeswijk (PRL84, 2000), Ota et al. (PLoS One, 2012), and Teramae et al. (PRL102, 2009), I am convinced that the theory developed here is mathematically wrong (in the sense of incomplete perturbation expansion).

3- Biological relevance

Here I assume the theory is somehow correct due to unknown reasons for a LIF model in a specific parameter range and only discusses the manuscript's biological relevance. In response to my earlier concerns about the biological relevance, the authors provide an additional figure that shows the theory and simulations matches where the Coefficient of Variation (CV) is below 0.2-0.3. They also argued that this is a relevant range for the sensory systems.

I need to admit that the suggestion of "Chacron et al. 2003" in my previous report was imprecise. I am sorry. I had in mind Fig.1 in Chacron et al. (JNeurosci, July 15, 2001) that the CV is about 0.2. Because in the original submission and revision, the input CV is about 10^{-5}-10{^-3} (minimal noise). However, it is still unclear if the model's firing rate, when the ISIs CV is about 0.2, also matches with this data point by Chacron et al. (JNeurosci, July 15, 2001).

There is enormous evidence that the CV in sensory systems can be much higher in contrast to what the authors claimed. To mention only a few examples: Locust auditory receptor neurons' data shows CV is above 0.5 (Schaette et al., JNeurophysio, 2005). Interestingly, Schaette et al. also report serial correlations in the data and indicate its functional relevance. In the early fly visual system, neurons can also have a very high CV of about 1.9 (see Fig.2 in Steveninck et al., Science 21, 1997). The high variability in spiking statistics of olfactory receptor neurons (ORN) is widely well known (see Duchamp‐Viret et al. (JNeurobio 65, 2005), where it shows the CV range for ORN is 0.5-3.0). 

Therefore, the theory (if it is mathematically correct for some unknown reasons) is somewhat limited and cannot be applicable as general, as the authors claim in the abstract.

In summary, I believe this work offers no perspective as a theory to solve higher-order statistics of neuronal models since it misses a more in-depth insight into the phase reduction methods in noisy conditions. Therefore, I insist that this work is not suitable for PLoS Comp. Bio. on three grounds: (i) it lacks an overview of relevant earlier works. Some related results have already been reported previously, and the manuscript is silent about them. (ii) The biological relevance is limited. (iii) The theory seems to be mathematically incorrect.

**Have all data underlying the figures and results presented in the manuscript been provided?**

Reviewer #2: Yes

PLOS authors have the option to publish the peer review history of their article (what does this mean?). If published, this will include your full peer review and any attached files.

Reviewer #2: No

---

## [Decision Letter · Decision Letter 2]

28 Apr 2021

Dear Mr. Ramlow,

Thank you very much for submitting your manuscript "Interspike interval correlations in neuron models with adaptation and correlated noise" for consideration at PLOS Computational Biology.

As with all papers reviewed by the journal, your manuscript was reviewed by members of the editorial board and by several independent reviewers. In light of the reviews (below this email), we would like to invite the resubmission of a significantly-revised version that takes into account the reviewers' comments.

I have received reviews that address the current version of your manuscript and like the first round, they are mixed. Two ask for minor revisions. The third makes strong objections based on some of the material in the introduction in which the authors seem to imply that cortical variability is the same mechanism as single neuron variability, a point that is certainly not true. Thus, I think in the revised version of the paper, the authors should play this aspect down and furthermore convince the first reviewer that this work is relevant to networks of neurons. This may be a tall order and I hope the authors will consider it.

We cannot make any decision about publication until we have seen the revised manuscript and your response to the reviewers' comments. Your revised manuscript is also likely to be sent to reviewers for further evaluation.

Sincerely,

Bard Ermentrout

Associate Editor

PLOS Computational Biology

Samuel Gershman

Deputy Editor

PLOS Computational Biology

Reviewer's Responses to Questions

**Comments to the Authors:**

Reviewer #3: In “Interspike interval correlations in neuron models with adaptation and correlated noise” by Ramlow & Lindner, the authors develop a theory of how spiking models with thresholds and voltage (+aux vars) reset behave when a neuron is tonically firing subject to weak noise. In this and several of their prior papers, they analyze a quantity called the serial correlation coeff (SCC), which is the trial-average of interspike intervals (ISI) that are separated k apart (k=0 <=> variance of ISI). The novel contribution is they have extended prior work to now include 3 contributing effects: white noise, adaptation current and correlated (in-time) noise, the later 2 being the key because they are combined. Their prior works and other works only considered colored noise or adaptation separately. The work is certainly technically sound: taking integrals, manipulating various terms to get SCC, series expanding and working in the Fourier domain (as far as I can tell).

But I do not believe this paper warrants publication in PLOS CB. I write this with angst because I hold the works of these authors (Lindner) in the highest regard, here are reasons for my personal opinion.

Abstract: “.. computational studies have identified spike-frequency adaptation and correlated noise as the two main mechanisms..”

The authors argument that rich statistical spiking structures exist and need to be explained is valid, but I find it very hard to believe that a single cell setting (or statistically identical population) can realistically model spiking data — I’m not sure what specifics they have in mind but the paper reads as if these methods apply in general. Aren’t the network structures (coupling, balanced inhibition, prob of connections) and changes in state (top-down modulation) among the key driving forces that result in the rich spiking dynamics? Is there experimental evidence to show that an uncoupled single neuron endowed with adaptation currents subject to white and correlated noise is the same as brain recordings?

Author Summary: “As these correlations can significantly improve information transmission and weak-signal detection, it is an important task to develop analytical approaches to these statistics for well-established computational models”.

This is not convincing. Why do we need analytics for this class of models specifically for coding (info trans & signal detection)? What specifically is it about the simulations that are not helpful for these coding questions? Is there an application of these formulas that could be tested experimentally (and/or can we use these formulas to suggest future experiments) that one would not have without simulations? I’m not specifically asking for actual data to validate the theory, but I feel like this is oversold.

I am impressed that even with all of their assumptions they can get particular excellent matches to the simulations. Does the assumption that the period is on average T^* (the same as the 0 noise case) really hold throughout all of these parameters? Is checking this consequential for the theory? I thought it would be easily violated with strongly correlated noise and/or strong adaptation.

Although these calculations are interesting to a subset of the compneuro community, I believe they are better suited for another journal (e.g., Physical Review Letters or Physical Review X, or many SIAM journals) where the specific absence of cortical function or application to a particular neural system would be better received by readers.

I can see the qualitative connections between serial ISI (scaled) for describing spike stat dynamics, but how is it related to spike counts that tell us something about coding? There is a blurb on lines 148-149 that says there is a connection between this and Fano Factor, and they cite Ref [21]— I assume Fig 5 in Ref [21] is the connection. I think [21] describe how larger or smaller values of serial ISI are related to smaller Fano Factor, which in that study is where maximal gain of discriminability occurs. This is important scientifically, but I believe the value of precise serial ISI calculations is pretty far removed from spike stats that effect coding/decoding.

How does the theory vs sims plots look for just the numerator of eq (1), < (T_i - <t_i>)(T_{i+k}-<t_{i+k})>? That would have been interesting to see if the net effect of the accuracy is because both num and denom are accurate or is something washed out in the normalization. I think the pieces are there (eq (34)).

I definitely liked the expositions around figs 4,5,6,7 (e.g., lines 185-206), they are technical but they handled the dynamics well. Fig 8 is also key to see when the weak noise starts to break down, although the deviations from simulations are small.

Minor:

- eq (2b) and line 77; is it that common to have the aux vars w_j reset upon spiking? I feel like that is very artificial, not commonly done in any biophysical models I have seen.

- line 342 “permits in most cases only numerical solutions” Don’t the formulas in eqs (3)—(6) require numerical calculations of integrals and various functions? How about the PRC Z (even in eqs (7), (8), (10)), and the adjoint method is mentioned in Methods section, which has to be solved numerically. I guess the authors mean they can ‘transparently’ see how parameters effect SCC.

- lines 411-412 “, we may consider ISI correlations for neurons connected in the large recurrent networks of the brain”, but the theory here assumes the firing rate is constant (line 473) and the noise is weak; what large recurrent networks in the brain have these assumptions?</t_{i+k})></t_i>

Reviewer #4: Understanding the non-renewal nature of neural spike trains is a problem of long-standing and historical interest in theoretical and systems neuroscience. In recent years researchers, Prof. Lindner’s group prominent among them, have devoted a great deal of effort to develop tools to link the mechanisms that make neural activity non-renewal to measurable statistics of that activity.

Ramlow and Lindner present calculate the serial interspike interval correlations for multi-dimensional integrate-and-fire neurons in a tonically firing regime with 1) spike-triggered adaptation, 2) weak white and 3) weak colored noise. While the impact of each of those three features on ISI correlations had been studied separately, their interplay has so far eluded description. By showing how the combination of these mechanisms can give rise to qualitatively different ISI serial correlation functions than either mechanism alone, this paper takes an appreciable step towards linking these theoretical studies with experiments.

The authors have convincingly rebutted the comments of reviewer 2 from the previous round and I agree with their rebuttal. I believe this paper is thus a solid theoretical contribution, meriting publication in Plos Comp. Biol.

Below, I list questions and comments in the hope they might improve the paper.

Major comments:

The authors described equation 3, their central theoretical result, as a linear combination of the isolated ISI correlations \\rho_{k, a} and \\rho_{k, \\eta}. But \\rho_{1, \\eta} appears in the coefficient A, and the product \\rho_{1, \\eta} \\rho_{1, a} appears in the coefficient C. So isn’t this a nonlinear combination of the isolated serial correlation coefficient functions? Describing this as just as a linear combination of \\rho_{k, a} and \\rho_{k, \\eta} seems to unfairly neglect its complexity.

Minor comments:

Throughout the paper the authors reference Type 1/ Type 2 neurons (and also Type 1/ Type 2 PRCs, which they describe). I for one had forgotten the definition of Type 1 / Type 2 neurons, and would appreciate a reminder the first time they are used in the paper.

Lines 89-92: This sentence is devoted to describing a variable defined for convenience in intermediate calculations in the methods section. u(t) doesn’t appear in the subsequent equations of the results section so talking about it here confused me. To unify the text with figure 1, I would suggest defining u here.

Equation 4d: The variable a^*, on the right-hand side of \\nu=…, is not defined here and the definition is hidden much later in the methods section, hard to find from this early point of the results.

Line 466: Does the adjoint method provide a route to calculate the phase response to perturbations in the auxiliary or adaptation variables?

Reviewer #5: 1) I follow the argument of the authors, that they cannot cite all related papers. To my understanding reviewer 2 had pointed out these specific papers to underline on his major critics (see 4 below)

2) Biological relevance, significance, novelty, readership.

Spike-triggered cellular adaptation is ubiquitous in spiking neurons across systems in vertebrate and invertebrate species. Equally, colored noise is evident in neurons. Both influence output spike/interval statistics. The serial correlation of inter-spike intervals is a wide-spread phenomenon and has implications for information transmission and response reliability, this functional relevance is nicely covered in the Discussion section of the manuscript.

The treatment of both, colored noise and SFA, and the derivation of serially correlated intervals is, to my knowledge, new. I think this is also understood by reviewer 2 who does not dispute the novelty of the manuscript. Describing the causal relation mathematically and with a claim of generality makes a significant contribution.

PLoS CB aims at a broad readership. The present MS is certainly heavy to digest, too heavy for most of a broad readership. However, I believe that theoretical manuscripts like this need to have their place in PLoS CB if the mathematically treated problem is of broader relevance, which is given here because adaptation and input noise are fundamental aspects of how single neurons compute in a network.

3) Limited range of CV

The authors show in their new Fig. 8 – in response to the reviewer’s request – and state in the reply that the theory “…predicts the SCC quantitatively up to CVs of about 0.2”. This clearly is a rather low value for central brain neurons, e.g. in cortex. However, the qualitative results for the shown cases of higher CVs are still valid. When discussing the limitations of their approach, the authors state clearly that they had to exclude “neurons that are subject to strong stochastic input such as emerging from synaptic background noise in recurrent neural networks…” The high CVs in e.g. cortical neurons is indeed a result of the synaptic input noise. I thus understand that many neurons that do show high CVs empirically are excluded as non-treatable with the present approach.

At this point of the Discussion I therefore suggest to add reference to the limitations in CV up to values of 0.2 in a similar manner as the authors did so in response to the reviewer’s point 3 (1st revision). The authors may indicate that higher values are typically observed in neuron types that are explicitly excluded here.

4) Completeness of perturbation expansion approach

The major critical point of reviewer 2 in my opinion focuses on the question whether the approach taken here is methodologically appropriate. Reviewer 2 suggests that the perturbation expansion approach is incomplete. He argues that a correct theory requires to consider perturbation in the orthogonal direction. I understood the reviewer’s argument; However, I could not resolve whether the authors are correct in their way when stating “our theory is correct in leading order of the perturbation amplitude”.

My suggestion: Possibly, and more prominently in the Discussion or less prominently in the Methods section, the authors could rephrase and shorten what they are stating in response to the reviewer (“2. Mathematical correctness”). This statement may then be disputed at any time and by anyone who would be able to proof that the approach taken here is incomplete; or confirm the correctness. That, in my opinion, would be a normal and desired course of scientific progress.

5) Generality

Another aspect raised by the reviewer is that of generality. The authors already made clear restrictions with respect to the applicability of their approach in the above-mentioned Discussion paragraph.

Still, I believe the authors could strengthen their manuscript if they demonstrate applicability to a conceptually different type of relevant neuron model, e.g. a conductance-based model, possibly the Wang-Buzsaki model? This would, of course, imply quite a bit of additional work and thus I want to make clear that this is meant as a constructive suggestion rather than a request.

**Have all data underlying the figures and results presented in the manuscript been provided?**

Reviewer #3: **No: **not freely available

Reviewer #4: None

Reviewer #5: Yes

PLOS authors have the option to publish the peer review history of their article (what does this mean?). If published, this will include your full peer review and any attached files.

Reviewer #3: No

Reviewer #4: No

Reviewer #5: No
---

## [Decision Letter · Decision Letter 3]

8 Jul 2021

Dear Mr. Ramlow,

We are pleased to inform you that your manuscript 'Interspike interval correlations in neuron models with adaptation and correlated noise' has been provisionally accepted for publication in PLOS Computational Biology.

Best regards,

Bard Ermentrout

Associate Editor

PLOS Computational Biology

Samuel Gershman

Deputy Editor

PLOS Computational Biology

Reviewer's Responses to Questions

**Comments to the Authors:**

Reviewer #3: Thanks for all the work and clarifications.

Reviewer #4: The authors have addressed my comments from the previous round.

Reviewer #5: The authors have satisfactorily addressed my comments in the course of the rebuttal process. The added statements in the Results section (p. 5) and in the Methods (p. 25) on the limitations of the authors’ approach caution the reader and clearly limit the claims. The reference to a mean driven regime as a limiting regime provides an intuitive understanding even for readers that cannot or prefer not to follow the mathematical arguments. I agree with the authors that likely some cortical neurons operate in a mean-driven regime. The additional application to the Traub-Miles model and the consistent match of theory and numerical simulation is convincing and broadens the scope of the MS.

Both, reviewer 3 and the authors have put extensive effort into their arguments and counter-arguments and, on a positive note, I think that this dispute shows dedication to the scientific cause.

As in my previous review statement, I repeat my opinion that at this stage only publication of this manuscript permits public disputation / support of the authors’ approach and conclusions, which fulfills the original function of publication in science.

The authors’ idea to publish the review discussion (reviews and responses) has its charm and would favor transparency. If I understand correctly, the authors have the option to proceed with publishing the "peer review history".

**Have the authors made all data and (if applicable) computational code underlying the findings in their manuscript fully available?**

Reviewer #3: Yes

Reviewer #4: None

Reviewer #5: None

PLOS authors have the option to publish the peer review history of their article (what does this mean?). If published, this will include your full peer review and any attached files.

Reviewer #3: No

Reviewer #4: No

Reviewer #5: **Yes: **Martin Nawrot

---

## [Editor Report · Acceptance letter]

20 Jul 2021

PCOMPBIOL-D-20-00911R3 

Interspike interval correlations in neuron models with adaptation and correlated noise

Dear Dr Ramlow,

I am pleased to inform you that your manuscript has been formally accepted for publication in PLOS Computational Biology. Your manuscript is now with our production department and you will be notified of the publication date in due course.

With kind regards,

Olena Szabo
